# Restoring protein glycosylation with GlycoShape

Callum M. Ives [1,4], Ojas Singh [1,4], Silvia D'Andrea[1], Carl A. Fogarty[1], Aoife M. Harbison[1], Akash Satheesan[2], Beatrice Tropea[2] & Elisa Fadda [3] ✉

Despite ground-breaking innovations in experimental structural biology and protein structure prediction techniques, capturing the structure of the glycans that functionalize proteins remains a challenge. Here we introduce GlycoShape (https://glycoshape.org), an open-access glycan structure database and toolbox designed to restore glycoproteins to their native and functional form in seconds. The GlycoShape database counts over 500 unique glycans so far, covering the human glycome and augmented by elements from a wide range of organisms, obtained from 1 ms of cumulative sampling from molecular dynamics simulations. These structures can be linked to proteins with a robust algorithm named Re-Glyco, directly compatible with structural data in open-access repositories, such as the Research Collaboratory for Structural Bioinformatics Protein Data Bank (RCSB PDB) and AlphaFold Protein Structure Database, or own. The quality, performance and broad applicability of GlycoShape is demonstrated by its ability to predict N-glycosylation occupancy, scoring a 93% agreement with experiment, based on screening all proteins in the PDB with a corresponding glycoproteomics profile, for a total of 4,259 N-glycosylation sequons.

The native fold of a protein determines its biological function, regulating mechanisms driving protein–protein and protein–ligand recognition, as well as binding and unbinding events in operational or physiological thermodynamic conditions. The chemical nature of the amino acids and their precise sequence are key determinants toward the correct folding and regulate both protein structural stability and dynamics upon folding. As a result of these crucial roles, protein sequence is strictly safeguarded by genetic encoding to guarantee reproducible biological function. In all forms of life, this seemingly rigid, template-driven paradigm is complemented by post-translational modifications (PTMs), which label proteins through the covalent addition of functional groups or of more complex molecular entities, leading the system to specific functions or biological pathways.

One of the most abundant and mysterious PTMs is glycosylation, which refers to the enzymatically controlled functionalization of biomolecules with complex carbohydrates, or glycans. Glycosylation constitutes a remarkably flexible biological strategy, allowing sequence and structure changes to occur on the fly, reflecting environmental conditions in different species and states of health and disease in which proteins are required to operate. It is estimated that about 3–4% of the human genome is devoted solely to encoding for mechanisms regulating protein glycosylation[1,2], underscoring its importance and ubiquity. Glycosylation can occur both co- and post-translationally through the formation of covalent bonds involving the hydroxyl groups of Ser and Thr sidechains, leading to O-glycans, or the amide nitrogen of the Asn sidechain, generating N-glycans[3]. Less common yet highly conserved[4] modifications include the functionalization at C2 of the Trp indole sidechain with mannose, known as C-mannosylation[1,5,6]. The structural complexity of glycans and of glycosylation patterns is unparalleled in nature, where sequences can range from one monosaccharide to hundreds of units linked through linear and branched arrangements. The glycosidic linkages connecting different monosaccharides confer a high degree of flexibility to the structures, which become largely invisible to experiments[7–9] even in cryogenic conditions. Indeed, glycans are often removed to allow for protein crystallization[9]. These difficulties in the structural characterization of sugars are further enhanced by the

[1]Department of Chemistry, Maynooth University, Maynooth, Ireland. [2]Hamilton Institute, Maynooth University, Maynooth, Ireland. [3]School of Biological Sciences, University of Southampton, Southampton, UK. [4]These authors contributed equally: Callum M. Ives, Ojas Singh. ✉e-mail: elisa.fadda@soton.ac.uk

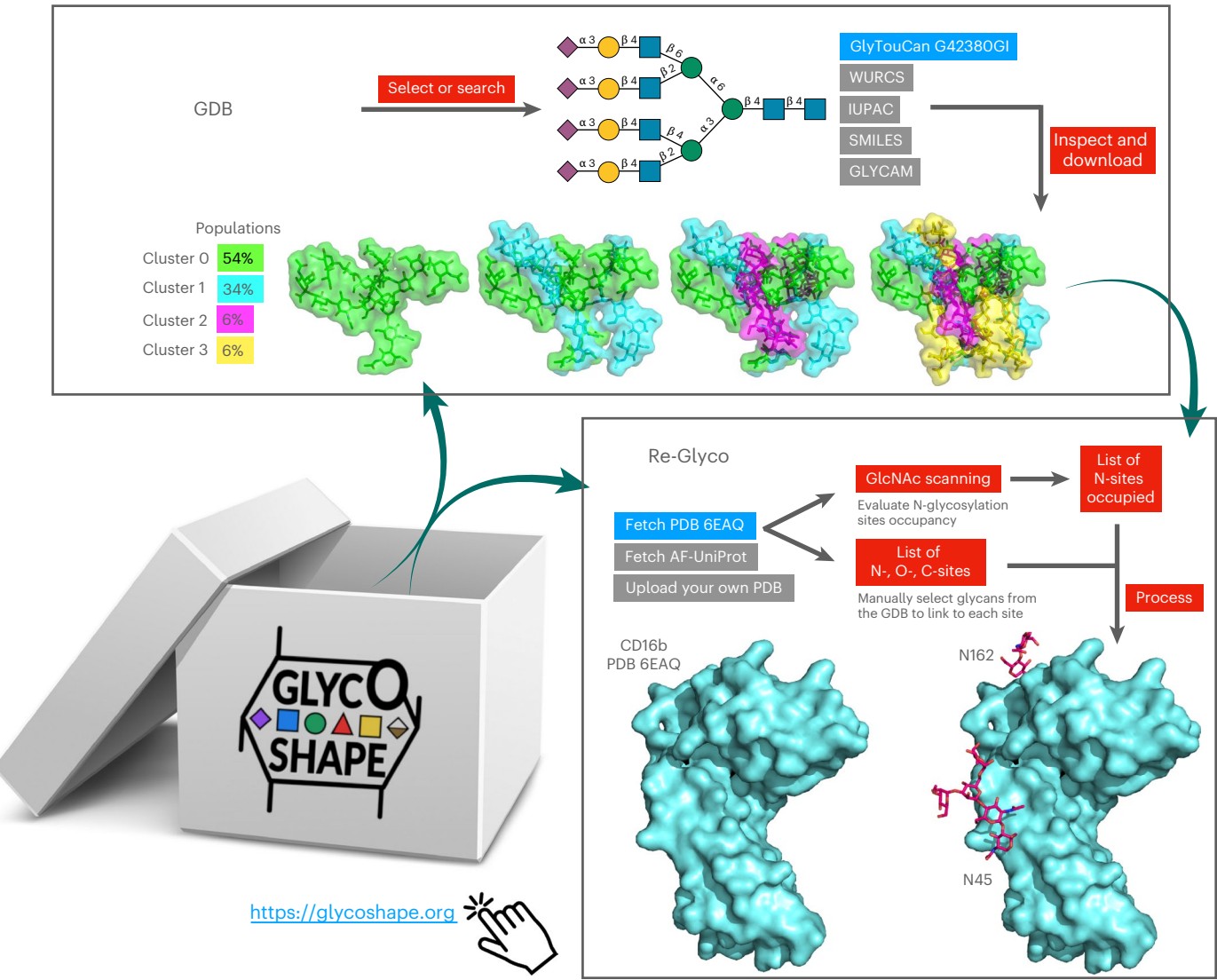

**Fig. 1 | Schematic representation of the GlycoShape workflow (https://glycoshape.org).** Top: the GlycoShape GDB is a repository of glycan 3D structures from 1 ms cumulative sampling through uncorrelated replicas of deterministic MD simulations. Structures can be searched by drawing a structure according to the Symbol Nomenclature for Glycans (SNFG) with the integrated SugarDrawer tool[83] or by searching by text in the International Union of Pure and Applied Chemistry (IUPAC), Web3 Unique Representation of Carbohydrate Structures (WURCS), Simplified Molecular Input Line Entry System (SMILES), GLYCAM and GlyTouCan formats. The GlyTouCan ID of the tetra-antennary complex N-glycan is shown in the example in blue. The successful search outputs general information on the glycan in the main tab and its 3D structure in the 'Structure' tab resulting from clustering analysis of the MD data with

the populations (weights) corresponding to each cluster. This information can be downloaded in PDB, CHARMM and GLYCAM formats. Bottom right: Re-Glyco allows users to rebuild the 3D structures of glycoproteins to the desired glycoforms by sourcing 3D glycan structures from the GDB and to predict N-glycosylation sites occupancy through the bespoke GlcNAc Scanning tool. As an example shown here and discussed in Results, the CD16b from PDB 6EAQ (in blue) is processed through the GlcNAc Scanning tool, which correctly predicts occupancy of the sequons at N45 and N16251. These sites can be N-glycosylated with a 'one-shot' approach, where the same N-glycan structure is chosen to occupy all, as shown in the example, or manually with a site-by-site approach, where the user can select a different N-glycan structure at each site.

absence of an encoding template, which determines varying degrees of macro- and microheterogeneity typical of all glycoconjugates[10–15].

The sparsity of information on glycan three-dimensional (3D) structures, occupation and identity at different sites contributes greatly to our lack of understanding of the many different functions that glycans and glycosylation play in biology. Yet, remarkable advances are continuously made in the development of new high-precision tools and techniques[16–26] that allow us to shed light on different aspects of the glycome. Within this framework, the potentials of glycobioinformatics[27–32] databases and resources and of high-performance computing molecular simulations[7,8] are extraordinary, especially within a context where information from multiple sources is required to decipher the

GlycoCode[33]. Here, we introduce GlycoShape (https://glycoshape.org) an open-access (OA) web-based platform designed to supplement the missing 3D structural information on glycans and glycoproteins, leveraging on over 1 ms of molecular dynamics (MD) simulations ran in dilute solution at standard conditions of temperature, pressure and salt concentration. A schematic overview of the GlycoShape workflow is shown in Fig. 1.

The GlycoShape Glycan Database (GDB) is continuously growing at a rate of approximately 30+ structures per week, and counts 534 unique glycans so far, predominantly sourced from the human glycome, but with selected examples of glycans structures, fragments and epitopes from other mammalian, invertebrate, plant, fungal and bacterial

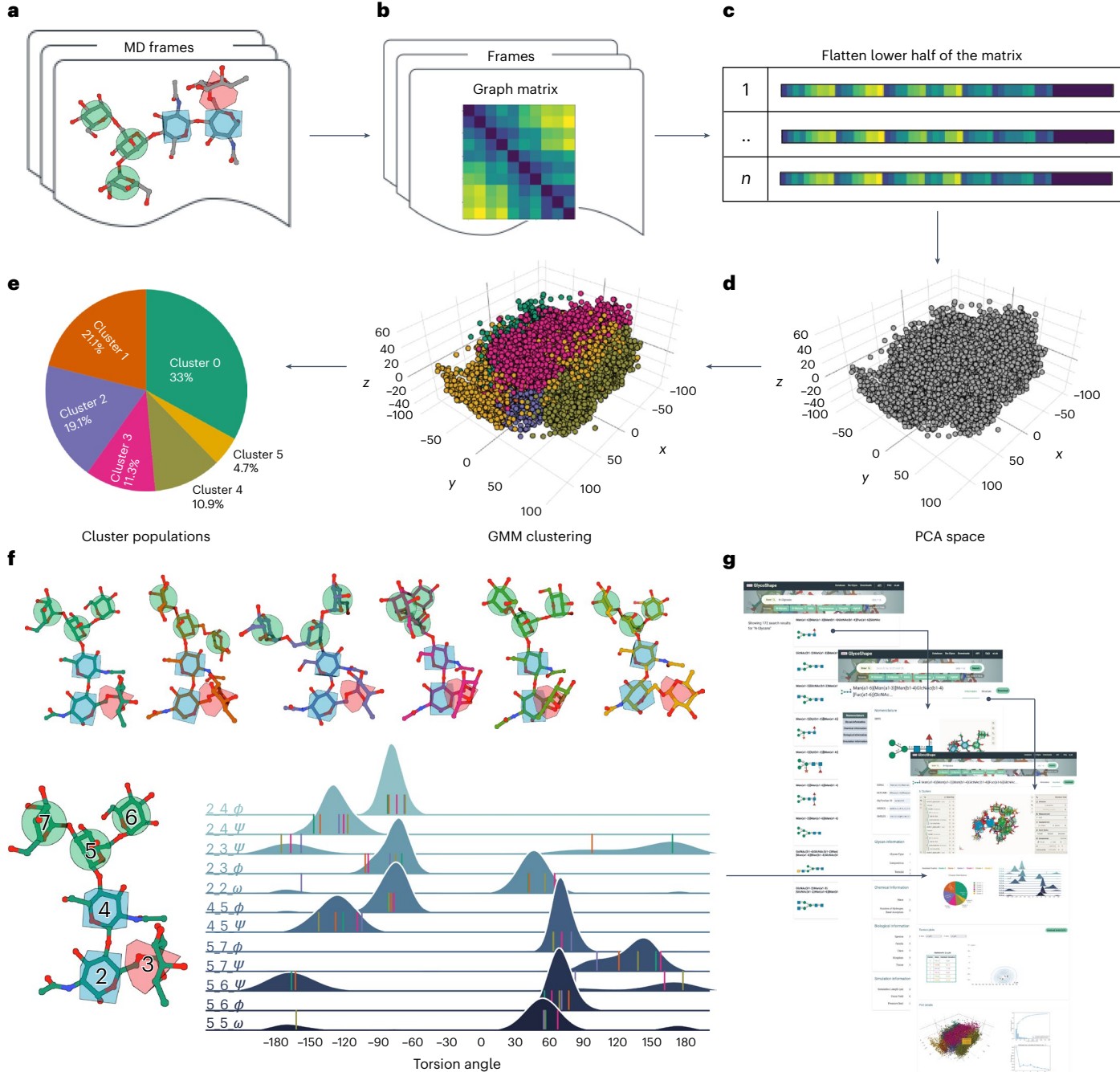

**Fig. 2 | Schematic overview of GAP used to build the GlycoShape GDB.**
**a**, Multiple uncorrelated replica MD simulations are performed for each glycan in the GDB, to comprehensively sample its structural dynamics. **b**,**c**, The resulting MD frames are then transformed into a graph matrix representation (**b**), simplified by flattening the lower half as a mono-dimensional array (**c**). **d**, This step enables a dimensionality reduction via PCA. **e**, These data are clustered by GMM, the results of which are displayed in terms of cluster distributions. **f**, Representative 3D structures for each cluster are selected on the basis of KDE maxima, along with comprehensive torsion angle profiles for the highest-populated clusters, showing the wide breadth of the conformational space covered by GAP. **g**, Structures derived from GAP are clearly presented on the GlycoShape GDB web platform, in addition to biological and chemical information.

species. Each glycan in the GDB is represented through distinct 3D conformers obtained from clustering analysis of the MD trajectories, with both α and β reducing ends and associated weights corresponding to the relative populations of the clusters during sampling. In GlycoShape, this 3D information can be used to rebuild glycoproteins to their functional, native state through a bespoke tool, named Re-Glyco (https://glycoshape.org/reglyco) (Fig. 1). To perform this task, the user can fetch protein structures directly from public repositories, namely, from the Research Collaboratory for Structural Bioinformatics Protein

Data Bank (RCSB PDB; https://www.rcsb.org/) with the corresponding PDB ID, and from the AlphaFold Protein Structure Database[34,35] (https://alphafold.ebi.ac.uk/) with the corresponding UniProt ID, or upload their own structure file in PDB format. Re-Glyco currently restores all types of C-, N- and O-glycosylation through a highly efficient genetic algorithm (GA) that minimizes a loss function accounting for steric hindrance. Based on the ability of this algorithm to evaluate accessible space and complementarity between the protein landscape and the glycan structure, Re-Glyco can also be used to predict the occupancy of

N-glycosylation sequons with a tool we named 'GlcNAc Scanning'. Testing GlcNAc Scanning on a subset of protein structures (739) from the PDB that have a corresponding glycoproteomics profile, as annotated in UniProt, shows a 93% agreement with experiment. Details on this test and data availability are included in Methods. In the next sections, we present and discuss the design, setup and features of the GlycoShape GDB and of Re-Glyco, and in Results we illustrate their use and performance through working examples, which also shows how GlycoShape can be used to improve and, where applicable, correct the underlying protein structure information. We then discuss the potentials of these tools for discovery and for advancing our understanding of the many roles of glycans in cell biology and in life science.

## The GlycoShape GDB

The conformational complexity of glycans is determined to a large extent by the inherent flexibility of the glycosidic linkages and can be further enhanced by transitions in the structure of the ring, or pucker[36–38], with probabilities depending on the type of monosaccharide[39]. Furthermore, unlike proteins that extend within a linear arrangement with peptide bonds structurally restrained by electron delocalization, glycans can be branched and each glycosidic linkage can potentially adopt two different anomeric orientations. The resulting combinatorial explosion of conformers would make the structure and dynamics of glycans virtually impossible to characterize. Yet, the glycans commonly found in eukaryotic glycoproteins are relatively small, counting 25 or fewer monosaccharides, with the exception of glycosaminoglycans, which for many reasons constitute a class of their own[40]. A reduced molecular size means that the number of degrees of freedom accessible to the corresponding glycan structures is relatively contained, compared with proteins or nucleic acids, making all-atom classical MD simulations a viable tool to fully characterize glycans' structure and dynamics[7,8] at infinite dilution, provided sufficient sampling is allowed.

Exhaustive sampling can be achieved by MD through enhanced sampling methods or by deterministic approaches, where the conformational space is explored in a user-controlled, stochastic manner, through the design of uncorrelated replicas set to cover all the energetically accessible degrees of freedom, which for most saccharides correspond to the glycosidic linkages with the highest flexibility[7,8]. The 3D structure and dynamics of each glycan in the GlycoShape database derives from this approach to MD sampling, where uncorrelated replicas of 500 ns are designed to sample different and accessible sections of the potential energy surface. For example, a glycan with three (1/2–6) linkages will be analyzed through sampling eight distinct conformations, as each linkage can access only two gauche (*gt* and *gg*) conformations of the three theoretically possible[7,8]. These eight uncorrelated MD replicas will generate a total of 4 μs of cumulative sampling data. The chemical nature of the glycosidic linkages determines that high energy barriers, difficult to overcome at room temperature, are uncommon, and in most cases, different conformers are seen to rapidly interconvert though this MD approach. These conformers are in equilibrium at room temperature, and their ensemble defines the glycan structure. Therefore, the dominant conformation of a glycan can change depending on how the environment stabilizes each conformer differently, shifting the equilibrium[41]. The GlycoShape platform provides all the representative structures of the conformers at equilibrium in solution with corresponding weights derived from the MD trajectories through our Glycan Analysis Pipeline (GAP) (Fig. 2).

In the GAP for each glycan, individual MD trajectories from the uncorrelated replicas are integrated into one dataset. Each frame of this cumulative trajectory is transformed into a graph matrix, that is, distance matrix, from which we derive a one-dimensional array by flattening the matrix's lower half (see Fig. 2b,c and Methods for details). This transformation captures all geometric data, setting the stage for a conformational landscape representation. To extract representative glycan structures, these data points are classified by clustering analysis.

Given the high dimensionality of the system, we first run a dimensionality reduction via principal component analysis (PCA), which leads to a 3D representation of the conformational landscape (Fig. 2d). These data are clustered using a Gaussian mixture model (GMM), where the optimal number of clusters is determined by the silhouette score, typically between two and ten clusters, often reaching an optimum at five (Fig. 2e). The selection of representative 3D structures for each cluster is not merely the centroid, which would correspond to a mean position lacking in conformational specificity. Instead, the most statistically significant conformation for each cluster is better defined through a kernel density estimation (KDE) analysis, which ensures a more accurate representation of the conformational space within each cluster (Fig. 2f). Based on GAP, users are able to access a wealth of structural information for each glycan structure, which includes not only representative conformational structures but also their population frequency during extensive MD sampling and detailed information on the $\phi$, $\psi$ and $\omega$ torsion angle values. In addition, the GlycoShape GDB complements this broad array of structural data with biological and chemical data of interest, in a clear and understandable format (Fig. 2g). More specifically, additional information includes alternative naming formats for each glycan, together with chemical and biological information directly sourced from different glycoinformatics repositories[27,42,43], as detailed in Supplementary Table 1. Further details on the design and implementation of the GDB are available in Methods.

## Rebuilding glycoproteins with Re-Glyco

Understanding the functional state of a protein requires restoring its co- and post-translational modifications (PTMs), which can affect its structural stability, dynamics and recognition/interaction ability. Re-Glyco is an algorithm implemented in GlycoShape, designed to restore glycosylation to protein 3D structures. The occupancy and specific type of glycosylation results from a complex equilibrium involving hundreds of glycosylhydrolases and glycosyltransferases operating through the secretory pathway[1,2,44], within a sequence with priorities still poorly understood. As the protein folds, the activities of these enzymes are regulated not only by the levels of expression but also by the physical accessibility of the glycosylation site[10,14,15,45,46]. Based on these considerations, restoring glycosylation onto a folded protein to produce representative 3D structures of a glycoprotein requires (1) access to exhaustive information on the type of glycan at each site[11,13,47], together with the conformational space accessible to those glycans[7,41], and (2) an algorithm able to assess the structural complementarity between these glycan 3D structures and the protein's rugged landscape surrounding the glycosylation site[41,45,46].

Re-Glyco sources glycans' 3D structures from the GlycoShape GDB. The conformational ensemble for each glycan is represented by the structures corresponding to the KDE-max of each cluster, obtained from the analysis of the MD simulations. Within the assumption of an exhaustive sampling, these 3D structures represent the conformational space accessible to the glycan at 300 K and 1 atm. As discussed in earlier work[41], the protein structure surrounding the glycosylation site modulates the conformational equilibrium corresponding to the glycan unlinked in solution, to favor the structure with optimal complementarity to the protein 3D landscape. For sites that are spatially accessible, the conformational equilibrium of the linked glycan is identical to the equilibrium corresponding to the unlinked glycan in solution[41].

The complementarity between the protein landscape surrounding glycosylation sites and the representative structures from each cluster is evaluated by Re-Glyco through a GA designed to minimize a loss function $F(P, G, \varphi, \psi)$ (Fig. 3a). This loss function accounts for steric hindrance in terms of linear distance between all the protein ($P$) atoms and all the glycan ($G$) atoms, except for the reducing end, with clashes counted within a minimum threshold value set to 1.7 Å, which corresponds to the van der Waals radius of a C atom. In the loss function $F(P, G, \varphi, \psi)$, $\varphi$ and $\psi$ are torsion angles values defining the conformation

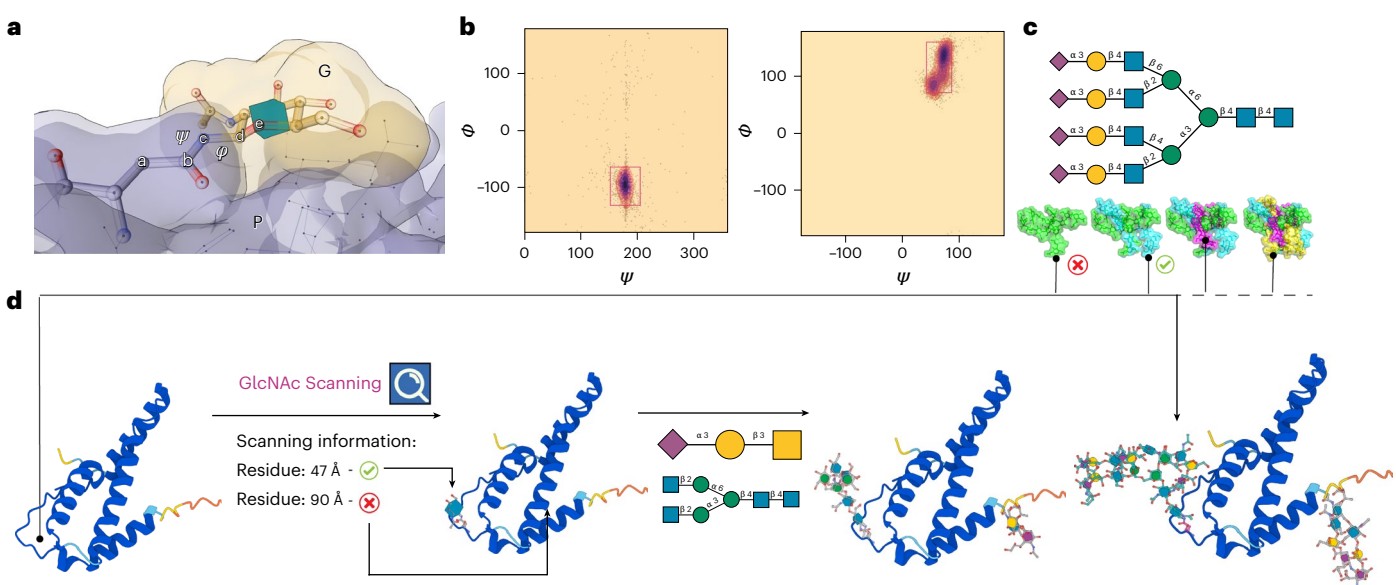

**Fig. 3 | Schematic overview of the Re-Glyco algorithm used to select and link glycan 3D structures from the GDB to a protein. a**, The definition of the $\varphi$ and $\psi$ torsion angles, with corresponding atoms labeled 'a' to 'e', determining the conformation of the linkage between the protein (P shown in gray) sidechain and the reducing end of the glycan (G shown in yellow). **b**, Heat maps showing preferential conformation of the $\varphi$, $\psi$ torsions between Asn-b-GlcNAc and Thr-a-GalNAc, with energy minima highlighted within red rectangles. **c**, Two-dimensional SNFG structure and (below) 3D structures of the tetra-antennary fully a3-sialylated N-glycan from the clustering analysis shown in Fig. 1. **d**, A schematic representation of the Re-Glyco workflow applied to the

reconstruction of human interleukin-5 (IL5; Uniprot P05113). In agreement with the annotation[84], GlcNAc Scanning identifies only the N47 sequon as potentially occupied. Accordingly, N47 can be functionalized with more elaborate structures through a 'one-shot' glycosylation, where also T22 can be functionalized with a sialylated core1 O-glycan. Highly complex glycosylation at N47 and alternative O-glycosylation structures can be selected by sourcing directly from the GDB through the Advanced (Site-by-Site) Glycosylation tool, as shown on the alternative IL5 glycoform on the righthand side. Molecular rendering with Mol* Viewer[85]; statistical analysis and heat maps created with matplotlib (https://matplotlib.org/).

of the bond to the protein (Fig. 3a). The values of $\varphi$ and $\psi$ have been sourced from Privateer[48] where available, or obtained from MD simulations ran specifically to complete the $\varphi$, $\psi$ dataset. The Re-Glyco workflow starts by testing the highest-populated glycan structure, namely, cluster 0 (G0), searching through the predetermined values of $\phi$ and $\psi$ torsions. Re-Glyco minimizes the loss function $F(P, G, \phi, \psi)$ over a population of 128 torsion sets ($\varphi, \psi$) across eight generations. If clashes persist, the search for $P-G0$ is aborted and Re-Glyco shifts to testing the second-highest-populated conformer from the GlycoShape GDB, that is, cluster 1 (G1), and so on. If none of the available glycan conformations fits the protein site, Re-Glyco progresses by trying to solve the $P-G(0-4/5)$ interaction that corresponds to the lowest steric hindrance through a 'wiggle' process. During the wiggle phase, torsion angle values are adjusted through random moves within a range of 10°, corresponding to the lowest range of the standard deviations commonly associated to the dynamics of glycosidic linkages at 300 K (refs. 7,8). This process is reiterated up to 40 times. The computational efficiency of this stochastic approach is vastly superior to any gradient-descent minimization algorithm, while providing more exhaustive sampling. Further details on the algorithm design and implementation are available in Methods. In addition to linking a glycan structure corresponding to the highest-populated conformer in the ensemble, Re-Glyco can also fit multiple frames selected from the MD trajectory with the 'Ensemble' module and calculate the corresponding solvent-accessible surface area (SASA) as an estimate of the space potentially available for protein–protein interactions[45], and to identify druggable sites.

The ability of Re-Glyco to resolve steric clashes can be used also to assess the potential occupancy of N-glycosylation sites through the 'GlcNAc Scanning' module (Fig. 3d). Where the N-glycosylation profile of a protein is not known, the user can choose to perform a GlcNAc Scanning, where Re-Glyco will try to fit a single GlcNAc monosaccharide into all the NXS/T sequons in the protein of choice with the protocol

described above. The process outputs a list of sequons that passed the test, marked with a simple 'yes' or 'no' label. The sequons labeled with 'yes' can be further N-glycosylated with the desired glycans by one-shot N-glycosylation, where the same N-glycan is chosen for all sites. If different types of glycan are required at each site, the user can manually choose the desired glycosylation with the Advanced (Site-by-Site) Glycosylation module. Alternatively, where information on the occupancy of the protein glycosylation sites is available through UniProt, the user can directly follow the given annotation and glycosylate the sites highlighted as occupied in one shot. Please note that this information in UniProt is often based on sequence analysis and may not correspond to the functional glycosylation state of the protein[49]. We advise users to always prioritize information supported by published references.

## Results

We present the results of two test cases where GlycoShape was used to restore the biologically active structure of specific glycoproteins. For this purpose, we chose to present particularly challenging cases, where the GDB and Re-Glyco are instrumental not only to rebuild glycosylation but also to flag important shortcomings in the protein structure, or in its use as a model for functional studies. Within this framework, we describe how the occurrence of 'false positives' and 'false negatives' can lead the user to refine the structural data, either own or sourced from a repository. As a general metric, the Re-Glyco GlcNAc Scanning tool predicts correctly 93% of the N-glycosylation sites, based on 739 proteins for which experimental structures are available and with N-glycosylation sequons annotated in UniProt as verified by experiments.

### Reglycosylation of CD16b

As a challenging working example, here we present the performance of Re-Glyco in detecting and rebuilding the functional form

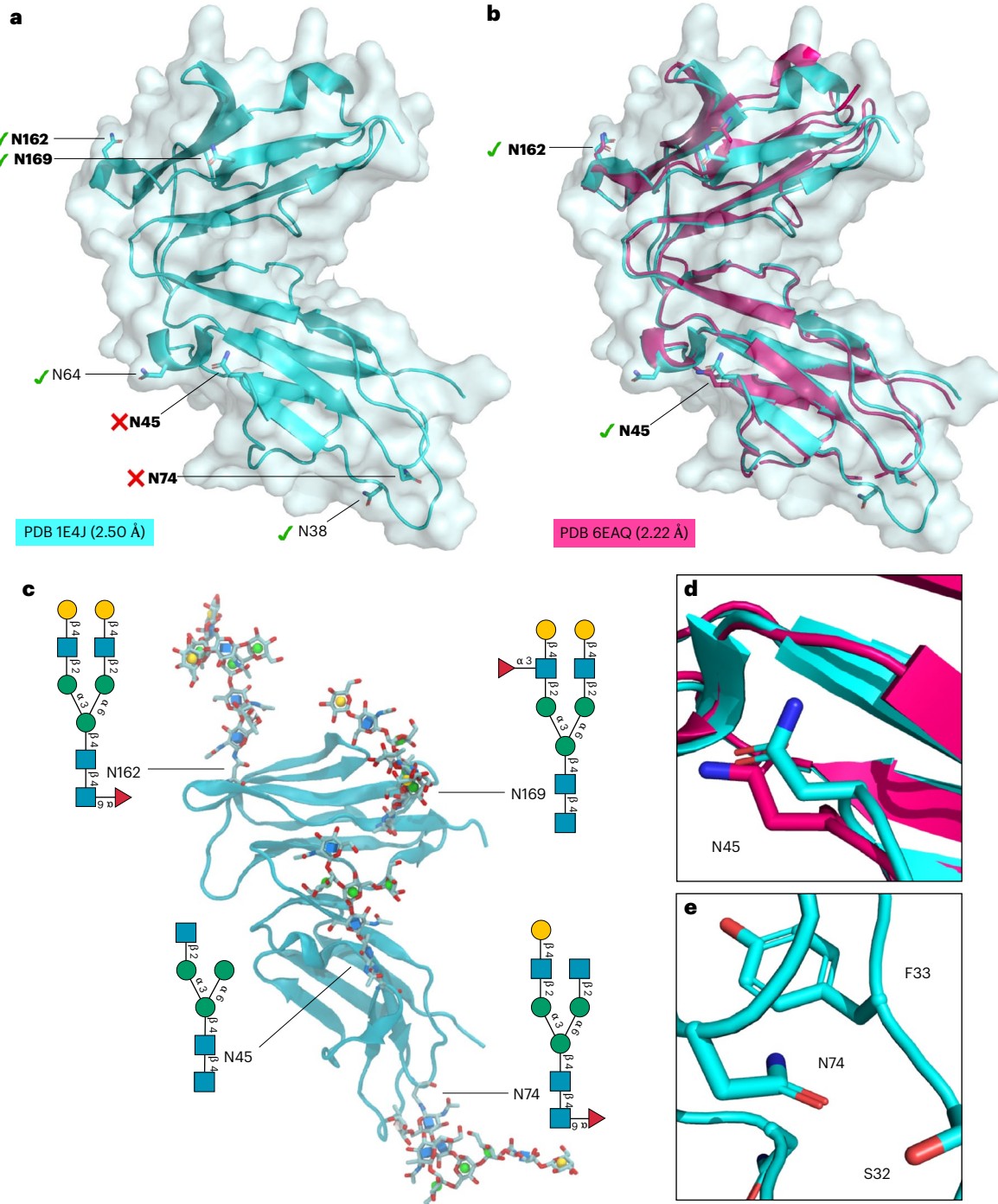

**Fig. 4 | Reglycosylation of CD16b. a**, The structure of the CD16b (PDB 1E4J, resolution 2.50 Å) with N-glycosylation sequons labeled. The bold labels indicate occupied sequons in neutrophil-bound CD16b[53]. The green check marks indicate the sequons predicted to be occupied by the Re-Glyco GlcNAc-Scanning tool, while the red cross marks indicate sequons that Re-Glyco deems unoccupied. **b**, The structural alignment of the CD16b (PDB 1E4J) in cyan with the CD16b (PDB 6EAQ, resolution 2.22 Å) in magenta. Both sequons occupied in PDB 6EAQ, namely, N45 and N162, are correctly predicted as occupied by Re-Glyco. **c**, The structure of the CD16b (PDB 1E4J) modified by swapping OD1 and ND2 coordinates and alternative rotameric orientation of the N74 sidechain

N-glycosylated by Re-Glyco with a different selection of N-glycan structures from the GlycoShape GDB at each site. **d**, A close-up view of the OD1 and ND2 orientation of N45 in the CD16b from PDB 1E4J (cyan) and from PDB 6EAQ (magenta). **e**, A close-up of the orientation of the N74 sidechain in the CD16b from PDB 1E4J. The distance between the CG of N45 and the CA of F33 is 4.7 Å. Rendering of the 3D structures in **a**, **b**, **d** and **e** and rotamer search performed with pymol (https://pymol.org/2/). Rendering of 3D structure in **c** with VMD[86] (https://www.ks.uiuc.edu/Research/vmd/) and N-glycan 2D structures with DrawGlycan-SNFG[87] (http://www.virtualglycome.org/DrawGlycan/).

of the Fc γ receptor III (CD16), the primary receptor responsible for antibody-dependent cell-mediated cytotoxicity[50–52] (Fig. 4). CD16 exists in two forms, CD16a and CD16b, that share 97% sequence identity. A crucial mutation from Gly to Asp at position 129 (G129D) in CD16b

was found to dramatically reduce binding affinity to the IgG-Fc relative to CD16a[51]. CD16a and CD16b share five of the six N-glycosylation sequons[53], and only one O-glycosylation site was detected in recombinant CD16a[54]. In agreement with this profile, we chose not to introduce

O-glycans in the reconstruction of CD16b, but the user can do that by selecting the site manually in the Advanced (Site-by-Site) Glycosylation tool and by adding the desired O-glycan structure from the GDB.

The crystal structure of the soluble CD16b in solution (PDB 1E4J) at 2.5 Å resolution, bears six N-glycosylation sequons (allotype NA2), namely, N38, N45, N64, N74, N162 and N169 (Fig. 4a). To note, Re-Glyco sources the residue numbers from the uploaded structure, so in the case of PDB 1E4J residues are indicated with a −3 shift, namely, N35, N42, N61, N71, N159 and N166. Here, we will use the former standard numbering above to avoid confusion. In the case of neutrophil-bound CD16b, an accurate N-glycosylation profile is available, not only indicating site occupancy but also the type of N-glycosylation found at each site[53]. According to these data[53], only four out of the six N-glycosylation sites are occupied in the neutrophil-bound human CD16b, namely, N45, N74, N162 and N169. The results of the GlcNAc Scanning of the CD16b structure from PDB 1E4J indicate that a single Asn-b-GlcNAc can fit at positions N38, N64, N162 and N169, missing two of the occupied N-glycosylation sites at N45 and N74 (two false negatives), and indicating as potentially occupied the unoccupied N38 and N64 (two false positives). As described below, a more in-depth analysis reveals the source of these apparent discrepancies, while alerting the user to a few features of the PDB 1E4J structure that make it incompatible with N-glycosylation.

False negatives in Re-Glyco often flag an orientation of the Asn sidechain that is unsuitable for N-glycosylation, which in the specific case of N45 depends on the assignment of the positions of ND2 versus OD1 (Fig. 4a). Indeed, in the presence of an N-glycan at N45, the orientation of the ND2 and OD1 is inverted, as confirmed by the structure of the CD16b in PDB 6EAQ (Fig. 4b,d), which was resolved with N-glycans at N45 and N162, while sequons at N38, N64, N74 and N169 were mutated to QXS/T[51]. The assignment of coordinates to ND2 and OD1 is largely arbitrary in structures obtained from X-ray crystallography, so in the case of a clash, Re-Glyco shows the link to the 'Swap' tool that allows the user to interchange the coordinates of ND2 with OD1 (https://glycoshape.org/swap). The modified PDB can be downloaded and screened again.

False negatives can also indicate an incompatible orientation of the whole Asn sidechain, as shown by the case of N74 (Fig. 4e). Indeed, in PDB 1E4J, the N74 sidechain is facing a loop section between S32 and F33, which clearly hinders the addition of a GlcNAc. No further support of the actual orientation of the N74 sidechain is available, as the whole region is not resolved in PDB 6EAQ (Fig. 4b), suggesting a high degree of conformational disorder. In this case, the clash can be solved by choosing an alternative orientation of the Asn sidechain by rotamer search and reloading the modified PDB structure. Re-Glyco does not provide an automatic rotamer search, as we think that modifying the underlying protein structure, even if to a minor degree, should not be done unless informed by expert knowledge or advice. Within this framework, the ability to retain structural information as default is crucial for the correct reconstruction of the glycosylation site. This is especially important for structural data obtained at high resolution, where the relative orientation of Asn sidechains, or of Ser/Thr sidechains for O-glycans, can be unequivocally assigned. Our approach is designed to preserve this information, which allows to restore the correct orientation of the glycan relative to the protein.

Regarding false positives, Re-Glyco can predict occupancy of unoccupied sequons when the protein structure is a monomer or a fragment of a more complex system, or, as in the case of CD16b, a transmembrane receptor severed from the membrane. When only part of the protein is processed through GlcNAc Scanning, N-glycosylation sites that are buried in the functional form appear as accessible. Accordingly, the predicted occupation of the highly accessible N38 (Fig. 4) is probably denied by its proximity to the membrane in the neutrophil-bound form of CD16b[53]. Indeed, in agreement with the prediction by Re-Glyco, N-glycosylation at N38 was detected in the recombinant receptor[53], which corresponds more closely to the structure we processed.

Analogously, the false positive at N64, where occupancy is suggested by Re-Glyco but not experimentally verified in neutrophil-bound CD16b[53], is not straightforward to rationalize. As shown in Fig. 4, N64 is highly accessible and, indeed, it is shown to be occupied with highly processed complex N-glycans in the soluble form of CD16b[53,55], suggesting again that the complexity of the environment of which the protein is part can significantly affect occupancy. To note, N64 is the only N-glycosylation sequon not conserved between CD16a and CD16b, and it is only present in the CD16b allotype NA2 (ref. 53). In summary, within the limitations of an isolated PDB structure representing a membrane-bound CD16b receptor and on the basis of the original assignment of the Asn sidechain orientation in the deposited PDB, Re-Glyco provides an accurate prediction of the N-glycosylation profile of the recombinant CD16b allotype NA2. The occupied sequons can be functionalized with N-glycans to match the neutrophil-bound CD16b glycoproteomics profile[53], as shown in Fig. 4c.

## Re-Glyco on AlphaFold structures

In earlier work[56], some of us and others explored the potentials of protein 3D structures derived from deep learning to be completed with the necessary PTMs. More specifically, we argued that protein structures from the AlphaFold Protein Structure Database (https://alphafold.ebi.ac.uk/) can be readily completed with nonprotein elements[56], as predicted features are often learned from structures of active biomolecules, which may contain cofactors, metals, ligands and often PTMs. In the case of glycosylation, while the structural information on the glycan is often incomplete and sometimes erroneous[9], we find that the structure of the aglycone obtained from machine learning (ML) is suitable for direct functionalization[56].

In this work, we investigated this point further, by testing 3,415 proteins from the AlphaFold Protein Structure Database with glycosylation sites annotated in Uniprot. In the reconstruction of the corresponding 12,789 N-glycosylation sites with a basic complex heptasaccharide A2 from the GDB (GlcNAc(b1-2)Man(a1-3)[GlcNAc(b1-2)Man(a1-6)]Man(b1-4)GlcNAc(b1-4)GlcNAc; GlyTouCan ID G88876JQ), Re-Glyco reports no clashes in 10,821 cases (85%) and unresolvable clashes in 1,968 cases (15%). We find that, when the clash occurs between the glycan and spatially neighboring loops with a predicted conformation ranked within the lowest and highest range of the predicted per-residue local distance difference test (pLDDT) scores[34,57,58], a clash resolution is generally achieved by the use of Re-Glyco alone (Fig. 5a). In case of clashes with loop regions predicted with a medium-range confidence score, namely, 90 > pLDDT > 65 (Fig. 5a), the resolution may require a direct (manual or otherwise) intervention, which often simply involves the selection of an alternative orientation of the sidechain of the aglycone or of the clashing residue, as discussed in the previous subsection. A pLDDT value below 70 is reported to indicate a lower degree of confidence in the AlphaFold 3D structure prediction[58], yet the results of a community-based exhaustive evaluation[59] suggest that the user should feel justified in selecting alternative sidechain orientations for residues with an associated pLDDT score <90.

A conformational search aimed at finding the optimal orientation of a residue sidechain may not be a strategy amenable to all users. To this end, we show here an example of how clashes encountered during the reconstruction of glycoproteins from the AlphaFold Protein Structure Database can be often resolved by using ColabFold[60], as shown in Fig. 5.

To illustrate this case, we selected the structure of the human exostosin-like 3 (EXTL3), one of the five exostosins[61] known to contribute to the synthesis of the heparan sulfate backbone in the Golgi[62]. EXTL3 not only has an AlphaFold structure (AF-O43909-F1), but it was also resolved by cryo-electron microscopy[63] (PDB 7AU2 and 7AUA) and by X-ray crystallography[64] (PDB 8OG1 and 8OG4). In Fig. 5, we show the structures of the EXTL3 monomers from PDB 8OG1 (Fig. 5b), from AlphaFold (Fig. 5c) and from ColabFold (Fig. 5d) within the silhouette

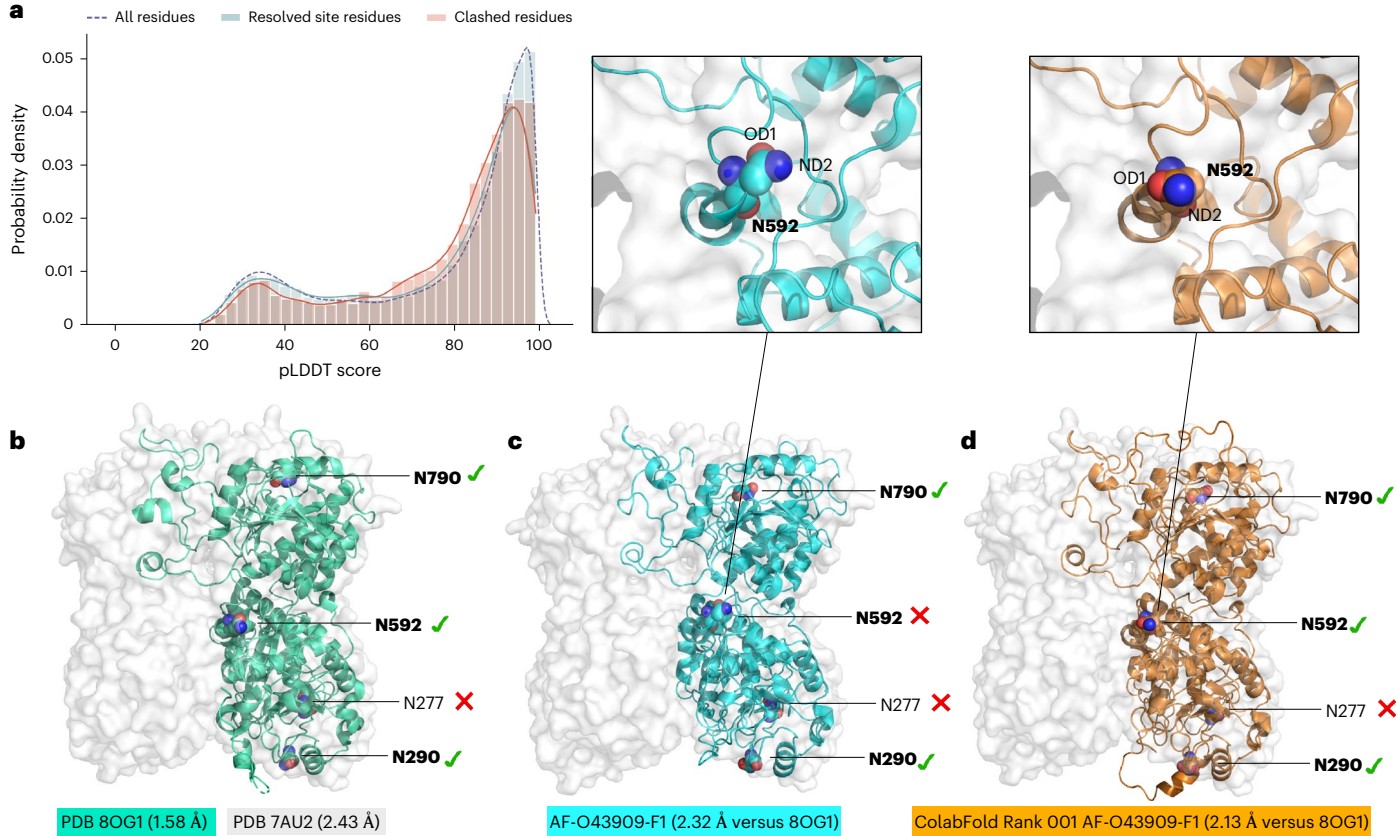

**Fig. 5 | Performance of Re-Glyco on AlphaFold structures. a**, A histogram analysis of the distribution of pLLDT scores of residues clashing during the GlcNAc Scanning of 3,415 proteins from the AlphaFold Protein Structure Database, with a total of 12,789 glycosylation sites annotated in Uniprot. The distribution of pLDDT values for all residues in the protein tested is shown with a dashed line. The distribution of the pLDDT values for the residues clashing with the GlcNAc during GlcNAc Scanning, where clashes were resolved by Re-Glyco is shown with powder blue histograms and a blue line. The distribution of the pLDDT values for the residues where clashing was not resolved by Re-Glyco and for the residues in the immediate vicinity (±2) is shown with rose histograms and a red line. **b**, The 3D structure of the EXTL3 monomer from PDB 8OG1 (green cartoons) represented within the homodimer from PDB 7AU2 (white surface). The resolution for each structure is shown in the labels. Asn residues within sequons are shown with van der Waals (vdW) spheres, where N atoms are in blue

and O atoms are in red. The N-glycosylation sequons known to be occupied are shown in bold. A green check mark indicates that the site is predicted to be occupied by Re-Glyco, while a red cross mark indicates that the site is predicted to be unoccupied by Re-Glyco owing to major steric clashing. **c**, Bottom: the 3D structure of the EXTL3 monomer from AlphaFold (AF-O43909-F1) shown in cyan. The lowest-confidence loops and termini are removed from the image for clarity. The all-atom RMSD versus PDB 8OG1 is shown in the label. Top: a close-up view of the sidechain orientation of Asn 592, where the clash with a spatially neighboring loop prevents functionalization. **d**, Bottom: the 3D structure of the EXTL3 monomer from ColabFold (Rank 001) shown in orange. Top: a close-up view of the sidechain orientation of Asn 592, showing the alternative orientation that allows for functionalization. Molecular representation with pymol (https://pymol.org/2), and statistical analysis and rendering created with matplotlib (https://matplotlib.org/).

of the homodimer based on the solvent-accessible surface from PDB 7AU2. N-glycosylation of human EXTL3 was experimentally confirmed[65] at sites N290, N592 and N790, while the sequon at N277 is unoccupied. GlcNAc Scanning with Re-Glyco of the experimental structures leads to the correct identification of all occupied sequons and also to the correct rejection of the sequon at N277. GlcNAc Scanning of the AlphaFold structure reports a false negative at N592 due to the wrong orientation of the Asn 592 sidechain (Fig. 5c). This clash is resolved by using one of the five structures provided by ColabFold[60], namely, Rank 001, which in this case gives an orientation of the N592 sidechain suitable for functionalization (Fig. 5d).

## Discussion

The reconstruction of the CD16b and of the EXTL3 structures highlights some of the features, performance and unique capabilities of GlycoShape that we trust will enable glycobiologists, structural biologists and the broader scientific community to easily restore glycoproteins to their native form. GlycoShape combines the power of extensive sampling from MD simulations to produce highly reliable insight into the structure and dynamics properties of biomolecules

that can be directly compared with experiments, enabling researchers to understand biological function at the molecular level of detail. We believe that these features, in combination with its computational efficiency, make GlycoShape unique and distinct from computational tools such as GlycoShield[66], designed to rapidly assess and quantify excluded volumes from glycan shielding, or from earlier work by Turupcu and Oostenbrink[67] proposing the reconstruction of glycans through free-energy calculations.

N-glycosylation is among the most common types of protein PTM, instrumental to protein folding and structural stability[3,44,68,69], as well as to mediating countless interactions between the cell and its environment in health and disease. N-glycans can also function as the first port of entry for pathogen infection[70] and are determinant to viral fitness and evasion[45,71-75]. Understanding the many roles of N-glycosylation requires the assessment of occupancy at each sequon and of the type of glycosylation at each occupied site. This information can be obtained by glycoproteomics, which so far remains the gold standard for protein glycoanalysis. Yet, glycoproteomics profiles are not available for all glycoproteins and require dedicated studies. The GlcNAc Scanning tool in Re-Glyco provides an additional, rapid screen

of the accessible volume at each N-glycosylation sequon and reports the potential occupation of the site by a single β-linked GlcNAc with a simple yes and no answer. This is in line with the key role of site accessibility as a determinant for N-glycosylation[10,11,14,45,76], along with other considerations, further supported by the correct identification of 93% of experimentally confirmed N-glycosylation sites in a subset of 739 proteins structures from the PDB. Re-Glyco provides the means to functionalize the sites predicted to be occupied with more elaborate glycan structures to match the desired profile. This selection should always be guided by glycoproteomics data, where available. Nevertheless, we are currently working on strategies to provide users with an informed choice of potential glycosylation profiles corresponding to protein expression in different cell lines.

In the case of O-glycans, prediction of site occupancy is still one of the most difficult challenges in glycobioinformatics. Functionalization with O-GalNAc glycans, which is the most abundant type of O-glycosylation of proteins[77] and that is inherently linked to their function[78,79], does not follow a precise amino acid sequence and is frequently encountered in highly flexible protein regions. Therefore, an approach based on sampling 3D space accessibility at potential O-glycosylation sites would not necessarily help. At this stage, users interested in reconstructing O-glycosylated sites with GlycoShape are advised to follow profiles obtained from glycoproteomics experiments and to rebuild the desired glycoform through a site-by-site approach in Re-Glyco. Nevertheless, a one-shot glycosylation is available also for O-GalNAc glycosylation, where the O-glycan selected as default is a sialylated core1 type.

In this Article, we have shown different cases where GlycoShape is not only useful to restore glycoprotein structures to their natural and functional form, but also how it provides an efficient strategy to identify minor flaws in the protein structure. Indeed, GlcNAc Scanning can indicate the position of the Asn OD1 and ND2 atoms that experimentally is unattainable and can also inform the choice of the Asn sidechain orientation. Where the protein structure requires a direct alteration to allow glycosylation with Re-Glyco to proceed, GlycoShape does not offer a rotamer search tool as we believe that direct modifications of the protein structure are not always justified and should be carefully guided by expert knowledge or advice. Yet, one can argue that for structures predicted by deep-learning rotamer optimization should always be allowed. Our analysis of a subset of glycosylated proteins from the AlphaFold Protein Structure Database shows that Re-Glyco finds unresolvable clashes more often with residues with a mid-range pLDDT score. We also show that these clashes can be resolved by using alternative outputs, which are easily obtained through ColabFold[60] or with the recently released AlphaFold 3 server[80]. Ultimately, the results in this work confirm an argument some of us presented earlier[56] as well as recent work from the developers of RoseTTAFold[81,82], that protein 3D structures from ML can be readily glycosylated and also used to predict glycosylation site occupancy, where this depends on 3D structure accessibility. For these reasons, we believe that a direct integration of GlycoShape within existing structural databases or ML workflows would greatly help users to include glycosylation in their studies and promote a better understanding of the many roles of glycans in the biology of health and disease and in life science.

## Online content

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

## Methods

### MD simulation protocol for free glycans

All glycans were built using the Carbohydrate Builder tool available in GLYCAM Web (https://glycam.org/cb/), carrying a hydroxyl group at the reducing end. Where flexible linkages are present in the structures, GLYCAM web allows users to select all (or selected) energetically accessible structures corresponding to the gauche–gauche (*gg*), gauche–trans (*gt*) or trans–gauche (*tg*) conformations of the linkage(s). For glycan structures that have only a single conformer, we ran simulations in triplicates of 500 ns each. For glycans that have two energetically accessible conformers, we ran two uncorrelated simulations (replicas) of 500 ns each. For glycans with three or more energetically accessible conformers, we ran a single 500 ns MD trajectory for each conformer. With this approach, every glycan structure in GlycoShape is sampled for a minimum of 1.5 µs, ensuring a sufficient and consistent sampling of the conformational landscape. Parameters sets were assigned with the tleap module of AmberTools, with carbohydrate atoms represented by the GLYCAM06j-1[88] force field and the TIP3P water model[89] for solvent molecules. Modifications and functional groups not present in the GLYCAM06j-1 force field, such as phospho groups found in nematoda or in bacterial monosaccharides, were represented by the general AMBER force field 2 (GAFF2)[90]. During system preparation, glycans were solvated in a water box with a minimum distance between 12 Å and 15 Å. Counterions (NaCl) were added to neutralize the total charge, where required, and to reach a concentration between 150 mM and 200 mM. The simulation conditions corresponding to each glycan are available on the Simulation Information tab on the GlycoShape website. All simulations were run using either AMBER 18[91] or GROMACS 2022.4[92] on resources provided by the Irish Centre for High-End Computing (ICHEC; www.ichec.ie), MeluXina (https://www.luxprovide.lu/meluxina/) and the Iridis computing cluster at the University of Southampton (https://www.southampton.ac.uk/isolutions/staff/iridis.page). For simulations run using AMBER 18, the energy of the system was minimized using the steepest descent algorithm, with the glycan heavy atoms restrained with a force constant of 5 kcal mol$^{-1}$ Å$^{-2}$. The system was then equilibrated in the NVT ensemble and heated in a two-step protocol, up to 100 K first, and then from 100 K to 300 K. The system was then equilibrated in the NPT ensemble to reach a 1 bar equilibrium pressure. The temperature was maintained at 300 K using Langevin dynamics with a collision frequency of 1 ps$^{-1}$. Pressure was maintained at 1 bar using isotropic position scaling with a Berendsen barostat with a pressure relaxation time of 2 ps. The particle mesh Ewald (PME) algorithm was used to treat long-range electrostatics with B-spline interpolation of order 4. Van der Waals interactions were truncated at 11 Å. All bonds to hydrogen atoms were constrained with the SHAKE algorithm, allowing the use of a 2 fs time step. For simulations run with GROMACS.2022.4 (GMX), the topology (.prm7) and structure (.rst7) files produced by tleap were converted to GMX (.top) and (.gro) files, respectively, with ACPYPE[93]. The energy of the system was minimized using 500k steps of steepest descent. The system was then equilibrated in the NVT ensemble to reach the equilibrium temperature of 300 K and then equilibrated in the NPT ensemble to reach the equilibrium pressure of 1 bar. The temperature was maintained at 300 K using Langevin dynamics with a collision frequency of 1 ps$^{-1}$, and the pressure was maintained at 1 bar using anisotropic position scaling with a Parrinello–Rahman barostat and a pressure relaxation time of 5 ps. Periodic boundary conditions were used throughout the simulations. The van der Waals interactions were truncated at 11 Å, and PME was used to treat long-range electrostatics. All bonds to hydrogen atoms were constrained with the LINCS algorithm to allow the use of a 2 fs time step.

### The GAP

The GAP algorithm is built on multiple processing components. We start by merging individual MD trajectories from the uncorrelated replicas into one dataset. During this step, all hydrogen atoms are removed to enhance computational efficiency. The conformations sampled through MD are represented by graph (distance) matrix. We built a function to first compute the flattened Euclidean distance matrix for a given set of data points. For a given dataset $X = \{x_1, x_2, ..., x_n\}$, where each $x_i$ indicates a point in the $d$-dimensional space ($d = 3$) in $\mathbf{R}^d$, the function computes the Euclidean distance matrix $G \in \mathbf{R}^{n \times n}$, where each element $g_{ij}$ is given by

$$g_{ij} = \|x_i - x_j\|_2 = \sqrt{\sum_{k=1}^{d} (x_{ik} - x_{jk})^2}. \tag{1}$$

The function then flattens the lower triangle of the matrix, excluding the diagonal, where the flattened array $G_f$ is defined as

$$G_f = \left\{ g_{21}, g_{31}, g_{41}, ..., g_{n1}, g_{32}, g_{42}, ..., g_{n2}, ..., g_{n(\frac{n-1}{2})} \right\}. \tag{2}$$

Each element in $G_f$ is obtained from the lower triangle of the distance matrix $G$. Finally, the function returns the transformed flattened array $G \in \mathbf{R}^{m \times p}$, where $p$ is the length of the flattened distance matrix $G_i$, and each row $i$ of $G$ contains the corresponding $G_i$.

The dimensionality of the array $G_f$ is reduced by PCA with the module sklearn.decomposition.PCA from sklearn[94]. We set the specified number of components 'dim' to 3 because it is the optimal value between the number of dimensions and efficiency of the algorithm, which decreases with the number of dimensions as points become more distant from each other in higher-dimensional spaces. The transformation can be represented as

$$T = G_f \times W, \tag{3}$$

where $T \in \mathbf{R}^{m \times \dim}$ is the transformed data matrix, and $W \in \mathbf{R}^{p \times \dim}$ is the matrix containing the top dimension principal components as columns. The function returns the transformed data matrix $T$ in the form of a DataFrame. In addition, the explained variance ratio and cumulative sum of the eigenvalues are calculated, which can be used to evaluate the explained variance of the PCA components as shown in Supplementary Fig. 1a.

We chose a GMM for 3D structures clustering as it presents distinct advantages over other clustering methods. GMM is a probabilistic approach that generates data from multiple Gaussian distributions. GaussianMixture from sklearn[94] is used to perform clustering with covariance_type = 'full', and random_state = 42 for reproducibility. The determination of the optimal number of clusters within GMM is guided by the silhouette score as shown in Supplementary Fig. 1b. This metric evaluates how similar an object is to its own cluster compared to other clusters, thus ensuring the most coherent and distinct grouping of the conformational states. In our analysis, we chose the maxima points of a KDE to serve as a proxy for the cluster centroids in a multidimensional feature space. This approach is particularly useful for elucidating the conformational landscape of molecular clusters, where traditional centroid-finding methods may not be applicable due to the irregularity and asymmetry of data distributions as shown in Supplementary Fig. 1c. To perform this analysis, we developed a strategy that comprises several steps. The KDE is defined as

$$\hat{p}(x) = \frac{1}{n \times h} \sum_{i=1}^{n} K\left(\frac{x - x_i}{h}\right). \tag{4}$$

Here, $K(\cdot)$ represents the Gaussian kernel function, expressed as

$$K(u) = \frac{1}{\sqrt{2\pi}} e^{-\frac{1}{2}u^2}, \tag{5}$$

and $h$ denotes the bandwidth, a parameter that influences the smoothness of the density estimation. To determine the optimal bandwidth, we employ a fivefold cross-validation scheme, where the dataset is partitioned into five subsets. The model is trained on four subsets and validated on the fifth, iteratively. The bandwidth that minimizes the validation error across all folds is selected. The search for the point nearest to the KDE's peak within each cluster begins by defining a negative KDE score function

$$s(x) = -\hat{p}(x). \tag{6}$$

Minimizing $s(x)$ equates to maximizing the KDE, thus pinpointing the density peak. For optimization, we use the L-BFGS-B algorithm, which allows for bound constraints, based on the minimum and maximum values across each dimension of the cluster's data points.

## The Re-Glyco glycoprotein builder

An overview of the Re-Glyco algorithm is presented as a flowchart shown in Supplementary Fig. 2. The following section contains key ideas involved in the development of the Re-Glyco tool. To restore protein glycosylation, using the glycan 3D structures from the GlycoShape database is necessary to know the correct geometry of the linkage (bond lengths, angles and torsions) between the reducing end of the sugar and the sidechain of the protein residue that needs to be functionalized. In the case of N-glycosylation, the torsion angles of the glycosidic linkages between the reducing GlcNAc and the Asn sidechain were obtained from Privateer[48], a software package for the automated conformational validation of sugar models. For O-glycosylation, we obtained these angles from MD simulations of exemplar proteins for each glycosylation type, as shown in Supplementary Table 1. Within this context, the topologies of syndecan-1 (ref. [95]) and the BAI-1 thrombospondin repeat 2 (TSR2) were built using the tleap module of AmberTools[96]. The AMBER 14SB force field[97] was used to model proteins and ions, the GLYCAM06j-1[88] force field was used to model glycans and the TIP3P water model[89] was used to model the solvent molecules. The structures of the 3-phosphoinositide-dependent protein kinase 1 (ref. [98]), the epidermal-like growth factor from coagulation factor IX[99,100] and dystroglycan 1 (ref. [101]) were made using the Solution Builder module of CHARMM-GUI[102]. The charged N- and C-terminal residues were neutralized by capping with an acetyl (ACE) and methyl-amido (NME) groups, respectively. The systems were solvated in a water box with a minimum distance of 12 Å, and ions were added to neutralize any system charges to a total concentration of 150 mM NaCl. The glycophorin A transmembrane system was made using the Membrane Builder module of CHARMM-GUI. The charged N- and C-terminal residues were neutralized by capping with ACE and NME groups, respectively. The protein was embedded in a 130 × 130 Å 1-palmitoyl-2-oleoyl-*sn*-glycero-3-phosphocholine membrane using the PPM v2.0 server[103], and ions were added to neutralize any system charges to a total concentration of 150 mM NaCl. The AMBER 14SB force field[97] was used to model proteins and ions, the GLYCAM06j-1[88] force field was used to model glycans, the Slipids[104] force field was used to model lipid molecules and the TIP3P water model[89] was used to model solvent molecules. For all soluble protein systems, MD simulations were run with AMBER 18 using the same system setup protocol described earlier for free glycans. For the transmembrane protein systems, the energy of the system was minimized using the steepest descent algorithm, with all atoms of the protein, lipid head groups and glycans restrained. The system was then equilibrated in the NVT ensemble, with the system gradually heated from 0 K to 100 K, and then from 100 K to 300 K. The system was then equilibrated in the NPT ensemble to maintain the pressure at 1 bar. Position restraints placed on the atoms of the protein, lipid head groups and glycans were gradually released during the equilibration process. The temperature was maintained at 300 K using Langevin dynamics with a collision frequency of 1 ps$^{-1}$, and the pressure was maintained at 1 bar

using semi-isotropic position scaling with a Berendsen barostat and a pressure relaxation time of 1 ps. Periodic boundary conditions were used throughout the simulations. The van der Waals interactions were truncated at 11 Å, and PME was used to treat long-range electrostatics with B-spline interpolation of order 4. The SHAKE algorithm was used to constrain all bonds containing hydrogen atoms and to allow the use of a 2 fs time step for all simulations.

Re-Glyco currently supports N-GlcNAcylation, O-GalNAcylation, O-GlcNAcylation, O-fucosylation, O-mannosylation, O-glucosylation, O-xylosylation and C-mannosylation.

The torsion angles ($\varphi$, $\psi$) regulating the geometry of linkage between protein ($P$) and glycan ($G$) and, ultimately, the glycan's orientation relative to the protein is shown in Supplementary Fig. 3. The [$\varphi_{range}$, $\psi_{range}$] values for all possible types of glycosylation are listed in Supplementary Table 2. The steric-based fitness function $F$ is defined as

$$F(P, G, \varphi, \psi) = \sum_{D_{ij} < 1.7} 200 \times \exp\left(D_{ij}^2\right) \tag{7}$$

and implemented on the basis of the algorithms shown in Supplementary Algorithms 1 and 2. In equation (7), $D_{ij}$ is defined as

$$D_{ij} = \sqrt{\sum_{k=1}^{d} \left(G'_{ik} - P_{jk}\right)^2},$$

the Euclidean distance between the $i$th element of $G'$ and the $j$th element of $P$. $G'$ is obtained after the operation rotate($G$, $\varphi$, $\psi$, $a$, $b$, $c$, $d$) that transforms the set $G$ into a new set $G'$ as shown in Supplementary Algorithm 3.

The algorithm we designed to identify glycosidic linkages within a molecular structure is shown in Supplementary Fig. 4. This algorithm functions by pinpointing torsion pairs that facilitate the molecule's ability to undergo conformational changes, and it is implemented using the networkX package[105] as shown in Supplementary Algorithms 4 and 5. Initially, the code identifies all the atoms forming cyclic structures within the molecule and isolates them for further examination. Subsequently, the algorithm scans through each atom in the noncyclic parts of the molecule, singling out those with only two bonds (that is, a degree of two). For these atoms, it identifies adjacent atoms and establishes torsion pairs, which are the quartets of atoms that define the axis of rotation for potential torsional movements. The collection of these torsion pairs effectively maps out all possible sites where the molecule can 'wiggle' or rotate around, thus providing alternate conformations of the glycan within a cluster.

The fitness function $F(P, G, \varphi, \psi)$ is optimized on the basis of a GA that iteratively improves a population of candidate solutions over multiple generations. The optimization starts with a population size of 128 pairs of parameters ($\varphi_{individual}$, $\psi_{individual}$), randomly initialized within their respective ranges [$\varphi_{range}$, $\psi_{range}$]. This procedure is described in Supplementary Algorithm 6. Each individual within this population is evaluated on the basis of the fitness function $F$, which measures how well the rotated conformation $G$ aligns with $P$. The algorithm then proceeds through eight generations during which the population is evolved by selecting the top 50% individuals (with the lowest fitness values) to serve as parents. These parents undergo crossover and mutation (0.2) operations to produce new offspring, which replaces the less fit individuals in the population. Throughout these generations, the algorithm seeks to minimize the fitness,

$$\text{Fitness}_{best} = \min(F(P, G, \varphi_{individual}, \psi_{individual})). \tag{8}$$

After eight generations, the algorithm identifies and returns the best individual ($\varphi_{best}$, $\psi_{best}$), which represents the optimal parameters that minimize steric interactions between the molecular conformations $P$ and $G$. The choice of 128 individuals to define the population size

and of eight generations is arbitrary and in our case corresponds to an optimal balance between exploration of the parameter space and computational efficiency.

### Re-Glyco Ensemble

Re-Glyco Ensemble can be used to fit multiple frames sourced from glycan MD simulations to enhance the diversity of glycan conformations and evaluate the exclusion volume on a protein surface due to the presence of a glycan. The process begins by randomly selecting 200 frames from the MD simulation of the free glycan of interest. Due to extensive sampling, we find that 200 frames is sufficient to account for the occluded space. For each selected frame, the $\varphi$ and $\psi$ torsion angles describing the linkage to the protein are generated using a uniform distribution based on the parameters specified in Supplementary Table 2. Steric compatibility is achieved through the computation of the fitness function $F$, shown in Supplementary Algorithm 1, for each frame and its respective $\varphi$ and $\psi$ angles. If any two atoms between $P$ and $G$ are found to be closer than 1.7 Å, corresponding to the van der Waals radius of a C atom, the frame is classed as clashing and is subsequently excluded from consideration. This process is repeated five times for each glycosylation site on the protein, ensuring a comprehensive and varied collection of conformations. The frames that successfully pass the steric clash test through these iterations are then aggregated into a multiframe PDB file. Subsequently, the SASA of the protein is calculated from the compiled multiframe PDB file using the gmx sasa module from GROMACS, for which we set the probe diameter to 0.14 nm and the number of dots per sphere to 15.

### Validation test of the GlcNAs Scanning algorithm and the AlphaFold2 test

The GlcNAc Scanning and AlphaFold2 test data are available in JavaScript Object Notation format in the GlycoShape GitHub repository. The test datasets for GlcNAc Scanning include structural information such as the UniProt ID, amino acid sequence and specific glycosylation sites with their resolved clashes, torsion angles and cluster assignments. Similarly, the AlphaFold2 test data provide insights into the computational time, clash resolution status, predicted glycosylation sites with their respective dihedral angles, and the pLDDT from AlphaFold's models. For data analysis and figures we used pandas, matplotlib[106] and seaborn[107] Python packages.

### Data cross-referencing

To complement the 3D structural data in the GDB, GlycoShape integrates information sources from other published resources and repositories to provide users with useful supporting information. More specifically, GlycoShape includes for each glycan all commonly used structure identifiers and naming formats, as well as chemical and biological information. This information is sourced from glycoinformatics resources, namely, GlyCosmos[42], GlyGen[27] and Glycowork[43], as shown in Supplementary Table 3. In GlycoShape glycans are identified primarily by their GlyTouCan IDs (https://glytoucan.org/) to make our GDB compatible with established glycobioinformatics resources and to allow for the development of a two-way accessibility between online resources, which is currently in development.

### Frontend and APIs

The frontend of the application was build and it is developed using React TypeScript, a powerful JavaScript library for building user interfaces with TypeScript (https://www.typescriptlang.org/), which adds static-type definitions to enhance code quality and understandability. For styling and theming, we used Chakra-UI (https://chakra-ui.com/), a simple, modular and accessible component library, to create a consistent and responsive design across the application. The frontend build process is managed using the npm@9.8.1 version (https://www.npmjs.com/), ensuring a streamlined and efficient compilation of the codebase. The search function of the GlycoShape website uses Levenshtein distance to calculate the differences and similarities between sequences and return the top results. This feature is augmented with the 'search-by-drawing-glycan' SugarDrawer tool[83], which simplifies the search for glycans, where the GlyTouCan ID or other identifiers are unknown or unavailable to the user. On the backend, GlycoShape leverages a set of Application Programming Interfaces (APIs) developed using Flask, a lightweight and powerful web framework for Python. These APIs are responsible for fetching and managing the data displayed on the website. To ensure reliable and continuous service, the APIs are served using Gunicorn, a Python WSGI HTTP Server behind a Nginx proxy server. This configuration offers enhanced performance, security and scalability, making GlycoShape a reliable tool for concurrent users.

The latest details and documentation of the stable APIs are available via GlycoShape API Documentation at https://glycoshape.org/api-docs, providing users and developers with comprehensive information on the API endpoints, usage and response formats, while facilitating the easy integration and utilization of GlycoShape's capabilities in various other glycobioinformatics software and resources.

### Reporting summary

Further information on research design is available in the Nature Portfolio Reporting Summary linked to this article.

## Data availability

The GDB is publicly accessible at https://glycoshape.org. The MD simulations used to curate the GDB are available for download at https://glycoshape.org/downloads.

## Code availability

The source code for the clustering methodology can be accessed via GitHub at https://github.com/Ojas-Singh/GlycanAnalysisPipeline. For the glycoprotein builder, the code is available via GitHub at https://github.com/Ojas-Singh/Re-Glyco. Code and scripts are also available open-access via Zenodo at https://doi.org/10.1101/2023.12.11.571101 (ref. 108).

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

## Acknowledgements

We thank all colleagues and friends who carefully tested the beta version of GlycoShape before its release, providing us with feedback, bug reports and crucial information that led us to finalize the current release. The Science Foundation of Ireland (SFI) Frontiers for the Future Programme is gratefully acknowledged for financial support of C.M.I.'s postdoctoral training, and of S.D.A.'s and O.S.'s postgraduate training (20/FFP-P/8809). The Science Foundation of Ireland (SFI) Centre for Research Training in Foundations of Data Science (www.data-science.ie) is gratefully acknowledged for financial support of A.S.'s and B.T.'s postgraduate training (18/CRT/6049). The opinions, findings and conclusions or recommendations expressed in this material are those of the author(s) and do not necessarily reflect the views of the Science Foundation of Ireland. C.A.F. acknowledges the Irish Research Council (IRC) for funding through the Government of Ireland Postgraduate Scholarship Programme (GOIPG/201912212). Please note that the funders had no role in study design, data collection and analysis, decision to publish or preparation of the manuscript. We gratefully acknowledge ORACLE for Research for the generous allocation of computational resources and R. Pitts, M. Riley and R. Kelly at ORACLE for their feedback, encouragement and support. The Irish Centre for High-End Computing (ICHEC) is gratefully acknowledged for generous allocation of computational resources. We acknowledge the use of the IRIDIS High Performance Computing Facility, and associated support services at the University of Southampton, in the completion of this work. E.F. gratefully acknowledges the contributions to the GDB derived from the work of Irene Cuxart Sanchez from C. Rovira Virgili's research group, during her placement in EF lab at Maynooth University. A.M.H. is currently a postdoctoral fellow at University College Dublin (UCD).

## Author contributions

C.M.I. developed and curated the GlycoShape GDB. O.S. developed the clustering analysis methodology, Re-Glyco and GlycoShape website. C.M.I., O.S., S.D.A., C.A.F., A.M.H., A.S. and B.T. ran all MD simulations used to build the GlycoShape GDB. E.F. designed and developed the GlycoShape project and overviewed its development. C.M.I., O.S. and E.F. analyzed the data and wrote the manuscript.

## Competing interests

The authors declare no competing interests.

## Additional information

**Correspondence and requests for materials** should be addressed to Elisa Fadda.

# Reporting Summary

## Statistics

For all statistical analyses, confirm that the following items are present in the figure legend, table legend, main text, or Methods section.

| n/a | Confirmed | |
|---|---|---|
| ☐ | ☒ | The exact sample size (*n*) for each experimental group/condition, given as a discrete number and unit of measurement |
| ☐ | ☒ | A statement on whether measurements were taken from distinct samples or whether the same sample was measured repeatedly |
| ☒ | ☐ | The statistical test(s) used AND whether they are one- or two-sided<br>*Only common tests should be described solely by name; describe more complex techniques in the Methods section.* |
| ☐ | ☒ | A description of all covariates tested |
| ☐ | ☒ | A description of any assumptions or corrections, such as tests of normality and adjustment for multiple comparisons |
| ☐ | ☒ | A full description of the statistical parameters including central tendency (e.g. means) or other basic estimates (e.g. regression coefficient) AND variation (e.g. standard deviation) or associated estimates of uncertainty (e.g. confidence intervals) |
| ☒ | ☐ | For null hypothesis testing, the test statistic (e.g. *F*, *t*, *r*) with confidence intervals, effect sizes, degrees of freedom and *P* value noted<br>*Give P values as exact values whenever suitable.* |
| ☒ | ☐ | For Bayesian analysis, information on the choice of priors and Markov chain Monte Carlo settings |
| ☐ | ☒ | For hierarchical and complex designs, identification of the appropriate level for tests and full reporting of outcomes |
| ☒ | ☐ | Estimates of effect sizes (e.g. Cohen's *d*, Pearson's *r*), indicating how they were calculated |

*Our web collection on statistics for biologists contains articles on many of the points above.*

## Software and code

Policy information about availability of computer code

| Data collection | Structural data was generated in our group by molecular dynamics simulations and analyzed in our lab. The simulations were conducted using either AMBER 18 or GROMACS 2022.4. All details on the methodology are presented in Supplementary Material. |
|---|---|
| Data analysis | All data analysis was done using python 3.10.6 with OA packages, e.g. MDAnalysis 2.7.0, pandas 2.2.1, numPy 1.26.4, scipy 1.12.0, plotly 5.9.0, bio-python 1.81, networkx 3.0, scikit-learn 1.2.2, matplotlib 3.8.3, seaborn 0.11.2 and other general python utilities. These were also embedded within data analysis tools created in our lab to address specific requirements, as described in the manuscript and in detail in Supplementary Material. All code is available OA in the https://github.com/Ojas-Singh/GlycoShape repository. Please also see the Code Availability section/statement in the manuscript. |

For manuscripts utilizing custom algorithms or software that are central to the research but not yet described in published literature, software must be made available to editors and reviewers. We strongly encourage code deposition in a community repository (e.g. GitHub). See the Nature Portfolio guidelines for submitting code & software for further information.

## Data

Policy information about availability of data

All manuscripts must include a data availability statement. This statement should provide the following information, where applicable:
- Accession codes, unique identifiers, or web links for publicly available datasets
- A description of any restrictions on data availability
- For clinical datasets or third party data, please ensure that the statement adheres to our policy

> The database is OA dowloadable from the https://glycoshape.org/downloads. All data is freely available, OA as indicated in the Data Availability statement in the manuscript

## Human research participants

Policy information about studies involving human research participants and Sex and Gender in Research.

| | |
|---|---|
| Reporting on sex and gender | *Use the terms sex (biological attribute) and gender (shaped by social and cultural circumstances) carefully in order to avoid confusing both terms. Indicate if findings apply to only one sex or gender; describe whether sex and gender were considered in study design whether sex and/or gender was determined based on self-reporting or assigned and methods used. Provide in the source data disaggregated sex and gender data where this information has been collected, and consent has been obtained for sharing of individual-level data; provide overall numbers in this Reporting Summary. Please state if this information has not been collected. Report sex- and gender-based analyses where performed, justify reasons for lack of sex- and gender-based analysis.* |
| Population characteristics | *Describe the covariate-relevant population characteristics of the human research participants (e.g. age, genotypic information, past and current diagnosis and treatment categories). If you filled out the behavioural & social sciences study design questions and have nothing to add here, write "See above."* |
| Recruitment | *Describe how participants were recruited. Outline any potential self-selection bias or other biases that may be present and how these are likely to impact results.* |
| Ethics oversight | *Identify the organization(s) that approved the study protocol.* |

Note that full information on the approval of the study protocol must also be provided in the manuscript.

# Field-specific reporting

Please select the one below that is the best fit for your research. If you are not sure, read the appropriate sections before making your selection.

☒ Life sciences  ☐ Behavioural & social sciences  ☐ Ecological, evolutionary & environmental sciences

For a reference copy of the document with all sections, see nature.com/documents/nr-reporting-summary-flat.pdf

# Life sciences study design

All studies must disclose on these points even when the disclosure is negative.

| | |
|---|---|
| Sample size | The sample sizes used in different tests include 435 unique glycan structures from MD in the Glycan Structure Database (GDB), 3,415 proteins from the AlphaFold Protein Structure Database from AI, 739 protein structures from experiment deposited in the PDB. The rationale used to determine sample size hinges on data availability, we used all experimental structures available, while the number of glycan structures determined covered all the known types of glycans in the human glycome with a sufficient degree of variability. The database, very much like the PDB continuously grows. |
| Data exclusions | no data were excluded |
| Replication | Each simulation was performed in uncorrelated replicates in function of the number of 1/2-6 linkages in the structure, with 2 replicas per linkage as described in detail in the manuscript and in Supplementary Material |
| Randomization | Randomization is not applicable in this context, not a clinical trial |
| Blinding | Blinding is not applicable in this context, not a clinical trial |

# Reporting for specific materials, systems and methods

We require information from authors about some types of materials, experimental systems and methods used in many studies. Here, indicate whether each material, system or method listed is relevant to your study. If you are not sure if a list item applies to your research, read the appropriate section before selecting a response.

## Materials & experimental systems

| n/a | Involved in the study |
|---|---|
| ☒ | ☐ Antibodies |
| ☒ | ☐ Eukaryotic cell lines |
| ☒ | ☐ Palaeontology and archaeology |
| ☒ | ☐ Animals and other organisms |
| ☒ | ☐ Clinical data |
| ☒ | ☐ Dual use research of concern |

## Methods

| n/a | Involved in the study |
|---|---|
| ☒ | ☐ ChIP-seq |
| ☒ | ☐ Flow cytometry |
| ☒ | ☐ MRI-based neuroimaging |

