## [Peer Review File · Nature Methods]

Restoring Protein Glycosylation with GlycoShape

Corresponding Author: Dr Elisa Fadda

Version 1:

Decision Letter:

27th Mar 2024

Dear Elisa,

Your Article, "Restoring Protein Glycosylation with GlycoShape", has now been seen by 3 reviewers. As you will see from their comments below, although the reviewers find your work of considerable potential interest, they have raised a few concerns. We are interested in the possibility of publishing your paper in Nature Methods, but would like to consider your response to these concerns before we reach a final decision on publication. We therefore invite you to revise your manuscript to address these concerns.

Link Redacted

We hope to receive your revised paper within 4-5 weeks. If you cannot send it within this time, please let us know. In this event, we will still be happy to reconsider your paper at a later date so long as nothing similar has been accepted for publication at Nature Methods or published elsewhere.

OPEN SCIENCE REQUIREMENTS

REPORTING SUMMARY AND EDITORIAL POLICY CHECKLISTS

DATA AVAILABILITY

All novel DNA and RNA sequencing data, protein sequences, genetic polymorphisms, linked genotype and phenotype data, gene expression data, macromolecular structures, and proteomics data must be deposited in a publicly accessible database, and accession codes and associated hyperlinks must be provided in the "Data Availability" section.

CODE AVAILABILITY

Please include a "Code Availability" subsection in the Online Methods which details how your custom code is made available. Only in rare cases (where code is not central to the main conclusions of the paper) is the statement "available upon request" allowed (and reasons should be specified).

For more information on our code sharing policy and requirements, please see: <https://www.nature.com/nature-research/editorial-policies/reporting-standards#availability-of-computer-code>

MATERIALS AVAILABILITY

SUPPLEMENTARY PROTOCOL

To help facilitate reproducibility and uptake of your method, we ask you to prepare a step-by-step Supplementary Protocol for the method described in this paper. We [encourage authors to share their step-by-step experimental protocols](https://www.nature.com/nature-research/editorial-policies/reporting-standards#protocols) on a protocol sharing platform of their choice and report the protocol DOI in the reference list. Nature Portfolio's Protocol Exchange is a free-to-use and open resource for protocols; protocols deposited in Protocol Exchange are citable and can be linked from the published article. More details can be found at www.nature.com/protocolexchange/about.

ORCID

Sincerely,
Arunima

Arunima Singh, Ph.D.
Senior Editor
Nature Methods

Reviewers' Comments:

Reviewer #1:

Remarks to the Author:

In this work entitled "Restoring Protein Glycosylation with GlycoShape," Callum, Singh et al. present an open access glycan structure database and toolbox, called GlycoShape, designed to restore glycoproteins in their native functional form. The GlycoShape Glycan Database (GDB) is a collection of glycan 3D conformations obtained through extensive molecular dynamics simulations. Users can search for hundreds (435) of N-glycan / O-glycan / GAG structures by drawing or using various text formats, among which IUPAC, WURCS, SMILES, GLYCAM. The database provides general information on each glycan and representative 3D conformations obtained from MD simulations. Glycan structures are downloadable in different formats. Glycoshape also includes several tools, among which "Re-Glyco" allows users to rebuild glycoprotein 3D structures by selecting glycoforms from GDB and predict N-glycosylation sites occupancy using another included tool named "GlcNAc Scanning tool". The user can fetch protein structures from RCSB PDB and AlphaFold, or provide their own PDB. Once performing the scanning, the user can decide whether to glycosylate in one shot with the same glycan or glycosylate site-by-site by simply picking the glycan from the glycoshape's GDB. Moreover, upon glycosylation, the user has the option to refine the glycosylated protein by selecting different glycan conformations from the MD ensemble of conformations. Everything is accessible on a webserver (<https://glycoshape.org>), but the user can also decide to install it and run it locally. The authors test the capabilities of their database and toolbox on two proteins, CD16b and EXTL3, highlighting the strengths, the limitations and some caveats.

I would like to remark that I am truly impressed by the amount of work done by the authors. Their glycan database is based on a staggering 1 ms MD sampling (cumulative), with multi microsecond long simulations performed for each glycan with either Gromacs or Amber using Glycam force field. For each glycan the authors provide extensive documentation, including simulations data and details on clustering. The effort done by the authors is absolutely phenomenal and holds potential for further improvements in the years to come. About the glycan database, the way it interfaces with the existing protein databases is somewhat smooth and makes it really easy to restore the glycosylated form of any available protein structure. The tools are easy to use, making the glycosylation of the protein very easy. The manuscript is elegantly presented in all its parts, the figures are sound, easy to read and follow along.

Overall, the Glycoshape glycan database and toolbox has all the credentials to become THE reference database / portal for glycans. As stated by the authors, and I could not agree more, Glycoshape will be widely used by "glycobiologists, structural biologists, and a broader scientific community". There are several databases to look for glycan structures (GlyYouCan for example), or to glycosylate proteins (CHARMM-GUI, among others) but this work is novel and provides a significant advancement with respect to all the other tools because it does everything. It is a database based on molecular dynamics simulations data that gives you access to ~500 different 3D glycan conformational ensembles (and not to only a single pre-

determined structure), it is integrated with PDB / AlphaFold, it restores glycosylation using the MD data solving potential clashes, it prints out PDBs and files that can be used by structural biologists or computational biophysicists. Also, you can choose or draw your glycans, and the interface is user-friendly, modern and elegant.

Although the authors present a few test cases, I extensively tried the toolbox myself, testing it against a very complicated protein, the SARS-CoV-2 spike protein, fetching the 6VSB cryo-EM structure. Now I can tell I wish I had this tool before! It took less than 10 minutes to complete the glycosylation of more than 50 sites with large M9 glycans (which is not how the spike is supposed to be glycosylated but only for the purpose of challenging the tools), and I was shocked by how easy it was to select the appropriate glycan for each site, and by the possibility of further refining the poses of the glycans using the MD data integrated in the database. Moreover, the program lets you know about the clashes that were not solved so that the user is aware of the possible issues when restoring the glycoform of a protein. The webserver is user friendly, easy to follow, with beautiful graphics. It is a pleasure and a treat to use.

For all these reasons, I totally endorse the publication in Nature Methods.

I have few minor suggestions that could improve the workflow / framework and make it accessible to an even broader community of users. I remark that these are only suggestions, NOT concerns/issues. I will divide them in suggestions that could be implemented for the purpose of publication in Nature Methods, and the ones that could be implemented in the future:

Suggestions for publication in Nature Methods:

1) Sometimes the system prompts errors even if the user is fetching the correct PDB ID or Uniprot ID. Then the screen updates anyway and the correct protein is indeed displayed, but there's some improvement to make it that would let the user know what is going on without leading them on to believe there's some mistake in the ID they are providing. I fetched 6VSB and I was getting the error alert red box while typing the ID, although I was then able to load the protein. I also tried with several Uniprot IDs, but it was somewhat not working properly (tried with: <https://www.uniprot.org/uniprotkb/P0DTC2/entry>). Maybe the authors could better specify when fetching a Uniprot ID is useful and when it should / should not be done.

2) It would be very useful to also add the Oxford notation, at least for the N-glycans (see here for example <https://www.ludger.com/docs/tables/ludger-n-glycan-nomenclature-table.pdf>). This is widely used and could simplify even more the search for the correct glycan.

3) When glycosylating with RE-GLYCO, a log file is printed to screen, informing the user about clashes: "Clash detected! Structure orientation for some spots are not glycan friendly." This is great, but when you have many sites, it becomes difficult to read all the lines. Therefore, I suggest adding a more compressed / readable summary where for each site it specifies only if there's a clash or not, and maybe (potentially) a suggestion on how to proceed (eg: use a different, smaller glycan, use SWAP, etc)

4) A direct button to access the SWAP tool would be helpful and improve the workflow. I could not find any button pointing to this tool from the main website. Disregard this if present and I did not find it. It seems that the SWAP tool is currently only accessible on a separate link (<https://glycoshape.org/swap>), which is somehow confusing. It would be useful to have it linked directly from the re-glyco window or from the main website.

5) When searching for glycans in the GDB, the user can download the pdb of the glycan with different atom scheme (CHARMM, Glycam, PDB): this is fantastic. I wish there was this option also when downloading the pdb of the glycosylated protein generated with re-glyco. Currently, the output of re-glyco only implements the PDB names for glycan residues, which makes it a little more complicated if one is going to use that structure for MD simulations with CHARMM36 or Glycam force fields. It would be terrific if the authors could also implement CHARMM36 and Glycam atom name scheme in the glycosylated protein generated with re-glyco. I note that CHARMM36 ff utilizes longer names for glycan residues (6 letters or more). Therefore, in this sense and if this will be implemented, some sort of remark should be printed to inform the user about this.

6) The authors may want to mention in the manuscript a few more tools that are available to perform glycosylation of protein, such as CHARMM-GUI and doGlycans. About CHARMM-GUI, the authors may want to consider to implement the CHARMM-GUI GRS code for drawing glycan structures. This is widely used in the simulation community.

Suggestions that could be implemented in the future:

-About the pdb that is generated by re-glyco, I also suggest the possibility of populating the segid/segname column of the PDB (4 characters) to assign a segid to each glycan. It could provide the user with an easier way to identify the different glycans when you have many glycans (i.e., 50 or more). Currently it is only based on the chain column, using letters or symbol.

-It would be fantastic to have some more detail printed out to screen while doing the scanning of the protein glycosites. As of now there's only this message printed: "It can take up to 5 minutes to process your request. Please wait. Please be advised that in the case of multiple users running simultaneously, your Re-Glyco job may take longer than expected." This is great but having some real time information printed on the go: "scanning sequon XXX..." would be very helpful.

-With respect to CHARMM, it would be very useful to also have a printout with all the CHARMM36 patches describing all the

glycosidic linkages within the glycan(s). These are required when setting up a glycosylated protein for MD simulations with NAMD and using charmm36 force field. An example of the patch is: "patch 13AA SEG1:resID SEG2:resID2", where 13AA is the name of the patch describing the glycosidic linkage between the following residues (identified by PRES in the charmm36 force field file). It could be beyond the scope, but it would be helpful to have this implemented somehow for the MD community.

Overall, I absolutely recommend publication in Nature Methods regardless the authors will be able to implement or not the suggestions listed above. Some of the suggested features may take longer time for implementation and I am sure the authors will take them into consideration anyway.

This is a brilliant manuscript and an innovative/revolutionary database/toolbox for restoring the glycosylated form of glycoproteins. I commend and thank the authors for such incredible work.

Reviewer #2:

Remarks to the Author:

The manuscript describes a resource to facilitate the creation of protein models that include glycan structures. The GlycoShape website contains a large database of three-dimensional structures of glycans. Furthermore, it provides tools to predict glycosylation based on structural feasibility and to build a wide variety of glycan structures onto a protein structure which has either been resolved experimentally, or was predicted. Considering the lack of three-dimensional structural data for glycosylation sites on proteins, this is an important contribution to the field of structural biology. The tool seems to work wonderfully, the manuscript is clearly written and reads well.

I would like to make the following comments:

1. For the representation of the distinct structures of glycans, the authors describe a 'glycan analysis pipeline'. It is not clear from the manuscript what the transformation 'into a graph matrix' actually means. What are the x_1, x_2, \dots, x_n in paragraph 3.1 of the SI? Cartesian coordinates, dihedral angles? It is stated in the main manuscript that a matrix is generated for every snapshot, so what do the differences g_{ij} relate to? Are these just intramolecular distances?
2. The re-glyco module is used to add the glycan structures to a structural model of a protein. The authors describe that it assesses the structural complementarity between the glycan 3D structures and the protein surface surrounding the glycan site. However, they may want to point out that this could be a relevant simplification, depending on the source of the protein structure: if the glycan is missing in the original protein structure because it was expressed in a host cell that lack the glycosylation machinery, or if the glycans were actively removed to facilitate the crystallization process, it may well be that the local protein conformation around the glycosylation site is very different than what it would look like in a glycosylated protein. Similarly, on page 14 of the manuscript, the authors refer to 'minor flaws' in the protein structure. The major flaw is the lack of glycans, the conformations that are observed under those conditions are not necessarily flawed if they do not accommodate a glycan.
3. Related to the previous point, in the examples the authors describe that N45 has a 'wrong assignment' of ND2 vs. OD1. While it is not uncommon that these atoms are assigned randomly in structures, I wonder how the authors know this is a 'wrong' assignment? Maybe this is the appropriate assignment for the aglycosylated and crystallized version of the protein? Similar considerations hold for the discussion of N592 in the second example, where an alternative pose from the ColabFold predictions resolved the clashes. This does not mean that for the aglycosylated protein the first pose is inappropriate.
4. The re-glyco algorithm includes an approach to find the interaction that corresponds to the lowest steric hindrance through a 'wiggle' process, in which torsional angles are sampled randomly within a 10 degree phase? Can the authors explain more clearly why they chose this approach and not just an energy minimization (possibly with a restriction on the maximum deviation of the dihedral angles) ?
5. Re-glyco essentially makes use of the pre-generated ensembles of glycan structures in solution to identify suitable instantiations for the protein that is being glycosylated. Such an approach is not new. Can the authors describe how re-glyco differs from the GlycoSHIELD approach (doi: 10.1016/j.cell.2024.01.034, previously available as doi: 10.1101/2021.08.04.455134) or from the approach described in doi: 10.1021/acs.jcim.7b00351 ?
6. If the selection of an alternative orientation of the sidechain of the aglycone or of the clashing residue may be needed to facilitate the presence of the glycan, would then a simple superposition of the backbone structure of the amino acid in the protein structure and in the single glycosylated asparagine in the reference ensemble not be more appropriate than the current approach through a selection via dihedral angles? Did the authors investigate such an approach? It would also largely address concerns 2 and 3, above.

Reviewer #3:

Remarks to the Author:

GlycoShape is truly a great software development to interface between such protein structure databases as PDB and AlphaFold to realise the effect that the ubiquitous translational modification of glycosylation makes to the structure of a protein. The authors have a true perspective of the importance of the heterogeneous glycosylation structures and their huge potential to vary the protein functions and the subsequent biology of all living organisms.

I am an experimentalist rather than a bioinformatician or software developer and therefore can only comment on its usefulness to the biologist – however I can see by reading the examples in this paper and trying the website out – that the potential that this knowledge brings is remarkable. The next step hopefully is the development of the capacity to look at

these 3D glycosylated proteins and model their interactions with their binding partners to determine the effect of different glycan epitope structures and the binding sites of their partner proteins.

It is exciting to see the integration of different glycoknowledge databases with ML protein structure databases with available reported experimental findings to make deductions of the real glycoprotein structure – of both protein and glycan. Database structures are good in that they show only structures where linkage is known and the SNFG colours and shapes are well used.. Also linking to GlyToucan identifiers IUPAC, GlyCam and programming languages is very good. Also like the Motif addition – can you search on this?

Comments

- It is a fascinating example of CD 16 – not only finding incorrect PDB structure but explaining why the N-linked sites were wrongly annotated when compared with experimental data. Need to justify better I think why O-site on CD16 was not included in the modelling?
 - The comment on future glycoproteomics - "with an informed choice of potential glycosylation profiles corresponding to protein expression in different cell lines." This may be tricky as the same protein will be glycosylated differently in different cell lines?
 - And "Rebuild glycoproteins to their functional native state " - not sure how you know that the glycans you added actually rebuild the functional aspects of the glycoprotein? A protein may not function for more reasons than steric hindrance e.g. glycan epitopes?
 - The heading and naming as "Restore" glycans – I understand that this means that a deglycosylated protein structure will have glycans added – but do we know this is actually restoring the in vivo structures? Perhaps another alternative word to "restore" – how about Reconstruct or Re-establish or.....
 - GlcNAc scan- "Where Re-Glyco will try to fit a single GlcNAc monosaccharide into all the NXS/T sequons in the protein" – will not the full N-linked structures also affect which sequon gets glycosylated? I accept that 93% of the structures you tried were correctly annotated but....?
 - A demonstration page would be useful on the webpage where the user is stepped through a real example – especially if the demo shows how to add your own structures to different sites – UniProt does not have all these possibilities as you have said!
 - Similarly in Sugar Drawer the "load Structure" tool is cool – though it wasn't clear to me how to add a single monosaccharide in a specific linkage – I eventually worked this out but again a demonstration example would be good?
 - Need to define "clusters " for the uninitiated as well as examples of what the plots on the structure pages are telling us?
- Overall the GlycoShape development is definitely needed for the field.

What I would say however is that I think the developers should work closely with an experimental group who trials the software and reports back on the pluses and minuses of the tool. This will ensure that it works in the hands of the less computationally informed and serves the experimental /biological community as well as it should.

Author Rebuttal letter:

Point-to-point Response to Reviewers

We would like to thank all three reviewers for their thorough analysis of our work and careful testing of GlycoShape and ReGlyco. We are truly humbled by their enthusiastic response and positive feedback and excited about the opportunity to share our resources with the community. We also appreciate all the suggestions for improving the manuscript and the website accessibility. We addressed all points raised in the response below.

Please note that the reviewers's comments are reported verbatim in italics, with key points highlighted in bold and underlined. Our answers are shown in normal text, highlighted in yellow.

Reviewer #1:

Remarks to the Author

In this work entitled "Restoring Protein Glycosylation with GlycoShape," Callum, Singh et al. present an open access glycan structure database and toolbox, called GlycoShape, designed to restore glycoproteins in their native functional form.

The GlycoShape Glycan Database (GDB) is a collection of glycan 3D conformations obtained through extensive molecular dynamics simulations. Users can search for hundreds (435) of N-glycan / O-glycan / GAG structures by drawing or using various text formats, among which IUPAC, WURCS, SMILES, GLYCAM. The database provides general information on each glycan and representative 3D conformations obtained from MD simulations. Glycan structures are downloadable in different formats. Glycoshape also includes several tools, among which "Re-Glyco" allows users to rebuild glycoprotein 3D structures by selecting glycoforms from GDB and predict N-glycosylation sites occupancy using another included tool named "GlcNAc Scanning tool". The user can fetch protein structures from RCSB PDB and AlphaFold, or provide their own PDB. Once performing the scanning, the user can decide whether to glycosylate in one shot with the same glycan or glycosylate site-by-site by simply picking the glycan from the glycoshape's GDB. Moreover, upon glycosylation, the user has the option to refine the glycosylated protein by selecting

different glycan conformations from the MD ensemble of conformations. Everything is accessible on a webserver (<https://glycoshape.org>), but the user can also decide to install it and run it locally. The authors test the capabilities of their database and toolbox on two proteins, CD16b and EXTL3, highlighting the strengths, the limitations and some caveats.

I would like to remark that I am truly impressed by the amount of work done by the authors. Their glycan database is based on a staggering 1 ms MD sampling (cumulative), with multi-microsecond long simulations performed for each glycan with either Gromacs or Amber using Glycam force field. For each glycan the authors provide extensive documentation, including simulation data and details on clustering. The effort done by the authors is absolutely phenomenal and holds potential for further improvements in the years to come. About the glycan database, the way it interfaces with the existing protein databases is somewhat smooth and makes it really easy to restore the glycosylated form of any available protein structure. The tools are easy to use, making the glycosylation of the protein very easy. The manuscript is elegantly presented in all its parts, the figures are sound, easy to read and follow along.

1

Overall, the Glycoshape glycan database and toolbox has all the credentials to become THE reference database / portal for glycans. As stated by the authors, and I could not agree more, Glycoshape will be widely used by glycobiologists, structural biologists, and a broader scientific community. There are several databases to look for glycan structures (GlyTouCan for example), or to glycosylate proteins (CHARMM-GUI, among others) but this is work is novel and provides a significant advancement with respect to all the other tools because it does everything. It is a database based on molecular dynamics simulation data that gives you access to ~500 different 3D glycan conformational ensembles (and not to only a single pre-determined structure), it is integrated with PDB / AlphaFold, it restores glycosylation using the MD data solving potential clashes, it prints out PDBs and files that can be used by structural biologists or computational biophysicists. Also, you can choose or draw your glycans, and the interface is user-friendly, modern and elegant.

Although the authors present a few test cases, I extensively tried the toolbox myself, testing it against a very complicated protein, the SARS-CoV-2 spike protein, fetching the 6VSB cryo-EM structure. Now I can tell I wish I had this tool before! It took less than 10 minutes to complete the glycosylation of more than 50 sites with large M9 glycans (which is not how the spike is supposed to be glycosylated but only for the purpose of challenging the tools), and I was shocked by how easy it was to select the appropriate glycan for each site, and by the possibility of further refining the poses of the glycans using the MD data integrated in the database. Moreover, the program lets you know about the clashes that were not solved so that the user is aware of the possible issues when restoring the glycoform of a protein. The webserver is user friendly, easy to follow, with beautiful graphics. It is a pleasure and a treat to use.

For all these reasons, I totally endorse the publication in Nature Methods.

We are delighted by the enthusiastic response from Reviewer 1 (Rev.1) and to hear that they found our method elegant, user-friendly, and worthy to become the reference database and toolbox for restoring the structure of glycoproteins. This was our aim all along and we are and will continue to strive towards keeping our resource up to the high standard indicated by Rev.1 for the benefit of the whole scientific community.

I have few minor suggestions that could improve the workflow / framework and make it accessible to an even broader community of users. I remark that these are only suggestions, NOT concerns/issues. I will divide them in suggestions that could be implemented for the purpose of publication in Nature Methods, and the ones that could be implemented in the future:

Suggestions for publication in Nature Methods:

1) Sometimes the system prompts errors even if the user is fetching the correct PDB ID or Uniprot ID. Then the screen updates anyway and the correct protein is indeed displayed, but there's some improvement to make it that would let the user know what is going on without leading them on to believe there's some mistake in the ID they are providing. I fetched 6VSB and I was getting the error alert red box while typing the ID, although I was then able to load the protein. I also tried with several Uniprot IDs, but it was somewhat not working properly (tried with: <https://www.uniprot.org/uniprotkb/P0DTC2/entry>). Maybe the

2

authors could better specify when fetching a Uniprot ID is useful and when it should / should not be done.

We thank Rev.1 for pointing this out. We fixed the glitch while typing the PDB/UniProt IDs. The Uniprot ID allows users to access the AF version of the whole protein structure, even when the protein corresponding to the same UniProt ID has structural data deposited in the PDB.

2) It would be very useful to also add the Oxford notation, at least for the N-glycans (see here for example <https://www.ludger.com/docs/tables/ludger-n-glycan-nomenclature-table.pdf>). This is widely used and could simplify even more the search for the correct glycan.

This is a great suggestion, as many do use the Oxford notation regularly for N-glycans, including some of us. We added the Oxford notation as a classification for the N-glycans, which can be now used for searches. I am not aware of a standard Oxford nomenclature for other glycans (not N-glycans), and we have not found any reference describing it. We will continue to investigate the issue and if a standard Oxford nomenclature for all glycans is available, will add the feature for all glycans in the database.

3) When glycosylating with RE-GLYCO, a log file is printed to screen, informing the user about clashes: âClash detected! Structure orientation for some spots are not glycan friendly.â This is great, but when you have many sites, it becomes difficult to read all the lines. Therefore, I suggest adding a more compressed / readable summary where for each site it specifies only if thereâs a clash or not, and maybe (potentially) a suggestion on how to proceed (eg: use a different, smaller glycan, use SWAP, etc)

We added a summary of the log, which (we agree) is more accessible, in addition to the full log that may still be useful to some users. The short log lists of glycosylation sites with indication of the presence/absence of a clash (simple yes/no labels).

4) A direct button to access the SWAP tool would be helpful and improve the workflow. I could not find any button pointing to this tool from the main website. Disregard this if present and I did not find it. It seems that the SWAP tool is currently only accessible on a separate link (<https://glycoshape.org/swap>), which is somehow confusing. It would be useful to have it linked directly from the re-glyco window or from the main website.

The integration of the Swap tool within the log was a bit problematic from the workflow perspective, but we did add a link to the Swap tool page to guide the users. We modified the text in the manuscript to reflect this change.

5) When searching for glycans in the GDB, the user can download the pdb of the glycan with different atom scheme (CHARMM, Glycam, PDB): this is fantastic. I wish there was this option also when downloading the pdb of the glycosylated protein generated with re-glyco. Currently, the output of re-glyco only implements the PDB names for glycan residues, which makes it a little more complicated if one is going to use that structure for MD simulations with CHARMM36 or Glycam force fields. It would be terrific if the authors could also implement CHARMM36 and Glycam atom name scheme in the glycosylated protein generated

3

with re-glyco. I note that CHARMM36 ff utilizes longer names for glycan residues (6 letters or more). Therefore, in this sense and if this will be implemented, some sort of remark should be printed to inform the user about this.

This is an excellent suggestion and indeed we are looking into this as part of the new release. Our rationale is that GlycoShape/ReGlyco is used by a broad range of researchers with different needs, expertises and expectations. We would like to make it easy for every user to find the information they need. To this end, for the next release, we are developing different âpackagesâ that will be accessible through the download button. One of these packages will be dedicated to users that will like to use the glycoprotein structure for MD simulations with GMX/Amber/CHARMM. In this package we will include the structure files with appropriate nomenclature and the topology file in the format desired. The addition of these features will require focused and careful work, so we opt to add this in the next release.

6) The authors may want to mention in the manuscript a few more tools that are available to perform glycosylation of protein, such as CHARMM-GUI and doGlycans. About CHARMM-GUI, the authors may want to consider to implement the CHARMM-GUI GRS code for drawing glycan structures. This is widely used in the simulation community.

The current implementation of the search algorithm is the result of extensive consultation with members of the broad glycobiology community, from glyco-bioinformaticians to structural glycobiologists, and researchers in glycoanalytics, cell glycobiology and carbohydrate chemistry. As biophysicists ourselves we are eager to add specialised tools to also support the MD community (see above). Yet, the implementation of something like the GRS is not possible as it is made to build glycan structures, while ReGlyco sources the whole structure directly from the GlycoShape GDB.

Suggestions that could be implemented in the future:

-About the pdb that is generated by re-glyco, I also suggest the possibility of populating the segid/segname column of the PDB (4 characters) to assign a segid to each glycan. It could provide the user with an easier way to identify the different glycans when you have many glycans (i.e., 50 or more). Currently it is only based on the chain column, using letters or symbol.

This is a very interesting suggestion, as we do understand very well the complexity of a PDB file that includes a high number of glycans. We will be required to keep the PDB files in their standard format for use in structural biology. Such format already contains link cards, which indicate which glycan goes where. Yet, in the `MD package` that will be available for download in the next release (see above), we will have more wiggle-room and add a specific SEGID. It is important also to note that secreted proteins are non nearly as densely glycosylated as viral proteins, so the SIGID information may be considered minor anyway.

-It would be fantastic to have some more detail printed out to screen while doing the scanning of the protein glycosites. As of now there's only this message printed: `It can take up to 5 minutes to process your request. Please wait. Please be advised that in the case of multiple users running simultaneously, your Re-Glyco job may take longer than expected.`

4

This is great but having some real time information printed on the go: `scanning sequon XXX` would be very helpful.

The addition of a `live report` is a bit tricky and we do not find it greatly informative in terms of our implementation of the algorithm. This is because delays in processing are not necessarily indicative of difficulties in fitting a glycan on the specific site, but more often are due to traffic on the server. For this reason we decided to hold off from introducing a live report now, yet we will continue to consider ways to do this for the next release.

-With respect to CHARMM, it would be very useful to also have a printout with all the CHARMM36 patches describing all the glycosidic linkages within the glycan(s). These are required when setting up a glycosylated protein for MD simulations with NAMD and using charmm36 force field. An example of the patch is: `patch 13AA SEG1:resID SEG2:resID2`, where 13AA is the name of the patch describing the glycosidic linkage between the following residues (identified by PRES in the charmm36 force field file). It could be beyond the scope, but it would be helpful to have this implemented somehow for the MD community.

This is a great suggestion and as such we have included it in our list of desirables for the `MD packages`. Please note that the PDBs have link cards already, as those are part of the standard format. These are useful indicators of what glycan goes where.

Overall, I absolutely recommend publication in Nature Methods regardless of whether the authors will be able to implement or not the suggestions listed above. Some of the suggested features may take longer time for implementation and I am sure the authors will take them into consideration anyway.

This is a brilliant manuscript and an innovative/revolutionary database/toolbox for restoring the glycosylated form of glycoproteins. I commend and thank the authors for such incredible work.

We would like to thank Rev.1 again for their analysis, feedback, and excellent suggestions, which will help make GlycoShape even more user-friendly and accessible to the community.

Reviewer #2:

Remarks to the Author:

The manuscript describes a resource to facilitate the creation of protein models that include glycan structures. The GlycoShape website contains a large database of three-dimensional structures of glycans. Furthermore, it provides tools to predict glycosylation based on structural feasibility and to build a wide variety of glycan structures onto a protein structure which has either been resolved experimentally, or was predicted. Considering the lack of three-dimensional structural data for glycosylation sites on proteins, this is an important contribution to the field of structural biology. The tool seems to work wonderfully, the manuscript is clearly written and reads well.

We are delighted to hear that Rev.2 found that GlycoShape makes an important contribution to the field of glycobiology, and that the tool works smoothly for the applications they chose to test it on.

5

I would like to make the following comments:

1. For the representation of the distinct structures of glycans, the authors describe a "glycan analysis pipeline". It is not clear from the manuscript what the transformation "into a graph matrix" actually means. What are the x_1, x_2, \dots, x_n in paragraph 3.1 of the SI? Cartesian coordinates, dihedral angles? It is stated in the main manuscript that a matrix is generated for every snapshot, so what do the differences g_{ij} relate to? Are these just intramolecular distances?

Graph matrices are distance matrices, so the variables are Euclidean distances. This was specified in the SI (old version) on page 7 paragraph 3.1. Nevertheless, we made sure in the new version of the SI that this is clear. We also updated the text in the manuscript on page 5 to make sure that there is no confusion, by indicating "... graph matrix, i.e. distance matrix, ..."

2. The re-glyco module is used to add the glycan structures to a structural model of a protein. The authors describe that it assesses the structural complementarity between the glycan 3D structures and the protein surface surrounding the glycan site. However, they may want to point out that this could be a relevant simplification, depending on the source of the protein structure: if the glycan is missing in the original protein structure because it was expressed in a host cell that lack the glycosylation machinery, or if the glycans were actively removed to facilitate the crystallization process, it may well be that the local protein conformation around the glycosylation site is very different than what it would look like in a glycosylated protein. Similarly, on page 14 of the manuscript, the authors refer to "minor flaws" in the protein structure. The major flaw is the lack of glycans, the conformations that are observed under those conditions are not necessarily flawed if they do not accommodate a glycan.

Rev.2 brings here a point that it would be difficult (if not impossible) to solve not only in a brief response, but also in general, as it is highly system dependent. The whole process of enzymatically removing glycans, or of expressing a protein in a cell line that does not perform glycosylation, is done under the well-founded assumption that this modification will not change the underlying structure of the protein, nor its function. As 70% of secreted proteins are glycosylated, this assumption underscores a large percentage of the proteins already deposited in the PDB. If the structural information on those systems was "inaccurate" or "incorrect", then a large fraction of structural biology and biophysics would most definitely suffer. So, in the following, we argue why we believe that the assumption above is well-founded and likely to hold. When a protein is de-glycosylated enzymatically, it is already folded into its stable, active state, so the modifications of structured regions (if any) induced by the removal of a glycan are likely minor. Changes to the structure and dynamics of "disordered" regions/loops surrounding, or allosterically connected, to the glycosylation site may indeed occur, yet those are not easily accountable from the static structure in the PDB, as those regions are rarely resolved. When a protein is expressed in a cell line that does not perform glycosylation, yet it folds to a stable functional structure, the glycan modifications are not likely determinant in "shaping" the native fold but are possibly affecting the folding thermodynamics/kinetics, or performing some other biological function. Restoring glycosylation to complement the structural information available helps to understand those functions. Ultimately, we believe that we made extremely clear in the manuscript that the

6

ability to restore glycosylation accurately hinges on the quality/accuracy of the underlying protein structural data, as this rationale applies to the broad field of structural biology and biophysics, not just to GlycoShape.

3. Related to the previous point, in the examples the authors describe that N45 has a "wrong assignment" of ND2 vs. OD1. While it is not uncommon that these atoms are assigned randomly in structures, I wonder how the authors know this is a "wrong" assignment? Maybe this is the appropriate assignment for the aglycosylated and crystallized version of the protein? Similar considerations hold for the discussion of N592 in the second example, where an alternative pose from the ColabFold predictions resolved the clashes. This does not mean that for the aglycosylated protein the first pose is inappropriate.

When the glycan is enzymatically removed the OD1 and ND2 positions can interconvert, provided that the local environment allows it. Yet, when occupancy of a specific N-glycosylation site is determined by glycoproteomics, the presence of the glycan most definitely will determine the ND2 vs. OD1 relative orientation, so in that sense, if the assigned positions contradict the presence of a glycan, those positions are indeed incorrect. Furthermore, it is important to underscore that the assignment of ND2 vs. OD1 coordinates from the electron density maps is largely arbitrary, unless the 3D structure is resolved at a resolution that allows the identification of hydrogen atoms, e.g. by neutron diffraction, so it is close to impossible to know that any assignment is right or wrong, unless an N-glycan is known to be at that site.

4. The re-glyco algorithm includes an approach to find the interaction that corresponds to the lowest steric hindrance through a "wiggle" process, in which torsional angles are sampled randomly within a 10 degree phase? Can the authors explain more clearly why they chose this approach and not just an energy minimization (possibly with a restriction on the maximum deviation of the dihedral angles)?

The reason we chose a random "wiggle" approach, rather than an energy minimisation is simply computational efficiency in combination with precision. The 10 degrees threshold is a low-bound of the normal standard deviation values corresponding to the glycosidic bonds torsion angles distribution during an MD simulation, so our stochastic approach is highly performing, in addition to its high computational efficiency.

5. Re-glyco essentially makes use of the pre-generated ensembles of glycan structures in solution to identify suitable instantiations for the protein that is being glycosylated. Such an approach is not new. Can the authors describe how re-glyco differs from the GlycoSHIELD approach (doi: 10.1016/j.cell.2024.01.034, previously available as doi: 10.1101/2021.08.04.455134) or from the approach described in doi: 10.1021/acs.jcim.7b00351?

The GlycoSHIELD and GlycoShape/Re-Glyco methods are radically different in both their design, philosophy and implementation. These differences originate from the purpose these two tools are made to fulfil. Re-Glyco makes use of equilibrium ensemble structures of glycans obtained through an exhaustive conformational search from deterministic MD simulations performed on uncorrelated replicas, produced based on a grid-search rationale. From the GAP analysis of the multi-microsecond MD simulations, we obtain representative

7

structures of the conformational clusters which, together with the corresponding populations, gives us all the possible conformers of each specific glycan in standard conditions. As the glycans conformational equilibrium depends only on its sequence, branching and stereochemistry, the Re-Glyco GA minimization of the carefully parameterised loss function, will find the structure(s) from the equilibrium ensemble that best complement the protein landscape.

Meanwhile, GlycoSHIELD is a tool built to provide information on the surface accessibility of densely glycosylated viral proteins, as its name suggests, in a similar fashion as the work from Besancon et al (DOI: 10.1016/j.ymeth.2019.07.010), and provides information on glycan-generated exclusion volumes, commonly used in pharmacophore searches for drug design. The structural details of the GlycoShield output are quite coarse, producing bond distances and angles that are often unphysical. Furthermore, the MD simulations used to produce the glycans conformers in GlycoSHIELD are very short, with the corresponding conformational sampling far from thermodynamic equilibrium. In our opinion, this is not a problem for small glycans, i.e. disaccharide up to specific tetrasaccharides, where conformational degrees of freedom are restricted and thus where the average structure does not diverge significantly from the structure obtained from carbohydrate builders (CHARMM-

GUI or GLYCAM). Yet this could be problematic in cases where glycans are larger and/or more flexible. This higher flexibility generates the "cloud" distribution that GlycoSHIELD uses to determine protein accessibility and such cloud may indeed spatially encompass the actual equilibrium ensemble, yet, unlike GlycoShape/Re-Glyco, GlycoSHIELD does not output a glycoprotein structure (see above), nor information about the glycans' complementarity to the protein structure, see more of this in the point raised below and in Bagdonas et al (10.1038/s41594-021-00680-9). We believe that information on protein-glycan complementarity is crucial to understand the functional roles of glycans, which do not just "shield" even in the case of viral proteins (see Casalino et al, <https://doi.org/10.1021/acscentsci.0c01056>). Information of protein-glycan complementarity provides key insight on the nature of the glycan itself, as we discussed in the case of CD16b in our manuscript. In summary, GlycoSHIELD and GlycoShape are profoundly different methods that serve radically different purposes and are only similar in that both approaches make use of MD as a tool to generate glycan 3D structures. Further to this, we would like to quote Rev.1 on this point "There are several databases to look for glycan structures (GlyTouCan for example), or to glycosylate proteins (CHARMM-GUI, among others) but this work is novel and provides a significant advancement with respect to all the other tools because it does everything."

The method described by Turupku and Oostenbrink in JCIM (2017) DOI: 10.1021/acs.jcim.7b00351, is different from both GlycoSHIELD, and GlycoShape. The approach represents a strategy to glycosylate glycoproteins in situ by reconstructing complex glycan structures from a (small) library of disaccharide and trisaccharide "blocks", with a conformational flexibility characterised by free energy (FE) calculations via umbrella sampling. Although aimed at the same result, ie. the generation of a glycoprotein, this approach is radically different from the one used by GlycoShape/Re-Glyco, also in the fact that the reconstruction in situ with Turupku and Oostenbrink's method is far from straightforward, potentially requiring further rounds of FE calculations, as it is also indicated in the original manuscript. Furthermore, the computational cost of the FE simulations is still prohibitive, as it was in 2017 when the paper was published, especially when the goal is to

8

cover to a sufficient degree the human glycome, or of any other glycome for that matter. Possibly this is the reason that has limited the development of the approach described and its application to single case studies.

6. If the selection of an alternative orientation of the sidechain of the aglycone or of the clashing residue may be needed to facilitate the presence of the glycan, would then a simple superposition of the backbone structure of the amino acid in the protein structure and in the single glycosylated asparagine in the reference ensemble not be more appropriate than the current approach through a selection via dihedral angles? Did the authors investigate such an approach? It would also largely address concerns 2 and 3, above.

As we discussed in manuscript, our approach is designed to make use of the structural information available obtained from experiments/predictions, and to produce a high quality glycoprotein structure. As we have shown both in the case of PDB and AF structures this requires refinement if the structural data is poor, yet is a huge strength for the correct reconstruction of the glycans when the structural data is good. At high resolution, the relative orientation of an Asn sidechain, or of a Ser/Thr sidechain for O-glycans, can be resolved very accurately, especially in cases where there is a glycan attached to it, yet partially or completely unresolved. For this reason, our approach is designed to preserve this important structural information, which is fundamental to determine the orientation of the glycan relative to the protein. Further to this, as discussed in the manuscript, Re-Glyco rebuilds the glycosidic linkage with an algorithm based on crystallographic parameters from Privateer (Agirre et al, <https://doi.org/10.1038/nsmb.3115>), completed by additional MD sampling for linkages not available. Stitching (or grafting) a sidechain linked to the glycan onto the protein backbone is possibly more computationally efficient, but it alters the conformation of the glycosylation site, so it would not work for us. Also the approach is likely to yield to unphysical conformations (see above), when not executed separately on single structures, which defies the computational speed. Meanwhile, this approach works just fine in the context of GlycoSHIELD, where the goal is to introduce an excluded volume on a protein surface, where the structural information on the centroid of that volume is not important.

Reviewer #3:

Remarks to the Author:

GlycoShape is truly a great software development to interface between such protein structure databases as PDB and AlphaFold to realise the effect that the ubiquitous translational modification of glycosylation makes to the structure of a protein. The authors

have a true perspective of the importance of the heterogeneous glycosylation structures and their huge potential to vary the protein functions and the subsequent biology of all living organisms.

I am an experimentalist rather than a bioinformatician or software developer and therefore can only comment on its usefulness to the biologist – however I can see by reading the examples in this paper and trying the website out – that the potential that this knowledge brings is remarkable. The next step hopefully is the development of the capacity to look at these 3D glycosylated proteins and model their interactions with their binding partners to determine the effect of different glycan epitope structures and the binding sites of their partner proteins.

9

We are truly excited about hearing such positive and enthusiastic feedback from Rev.3. We designed and built GlycoShape to support the work of experimentalists, as well as of computational scientists, and these comments clearly suggest that we are on the right path of doing that. Thank you.

It is exciting to see the integration of different glycoknowledge databases with ML protein structure databases with available reported experimental findings to make deductions of the real glycoprotein structure – of both protein and glycan.

Database structures are good in that they show only structures where linkage is known and the SNFG colours and shapes are well used.. Also linking to GlyToucan identifiers IUPAC, GlyCam and programming languages is very good. Also like the Motif addition – can you search on this?

Yes, search can be performed on any of those identifiers, including –motif– and now also through the Oxford nomenclature for N-glycans.

Comments

– It is a fascinating example of CD 16 – not only finding incorrect PDB structure but explaining why the N-linked sites were wrongly annotated when compared with experimental data. Need to justify better I think why O-site on CD16 was not included in the modelling?

The O-glycosylation site has been reported to be occupied in CD16a, not CD16b, which is the system discussed. We are sorry about the confusion and we added a comment in the manuscript to clarify this point. More specifically on page

– The comment on future glycoproteomics - –with an informed choice of potential glycosylation profiles corresponding to protein expression in different cell lines.– This may be tricky as the same protein will be glycosylated differently in different cell lines?

Absolutely correct and that is exactly the point we were trying to make. For future releases we are planning to give users an informed choice of what the glycosylation of that same protein may be upon expression in different cell lines. This is now work in progress.

– And –Rebuild glycoproteins to their functional native state – not sure how you know that the glycans you added actually rebuild the functional aspects of the glycoprotein? A protein may not function for more reasons than steric hindrance e.g. glycan epitopes?

Rev.3 is absolutely correct. The point we were trying to make in the manuscript is that a glycoprotein is only functional, or it does exist, only when it is glycosylated. Nevertheless, in order to avoid potential misunderstandings, we changed the text of the manuscript where we thought was most appropriate. For example, on page 13 we substituted –functional form– with –native form– of the protein.

– The heading and naming as –Restore– glycans – I understand that this means that a deglycosylated protein structure will have glycans added – but do we know this is actually restoring the in vivo structures? Perhaps another alternative word to –restore– – how about Reconstruct or Re-establish or –

10

We understand Rev.3 pondering as we have gone through the same debate ourselves. We finally (democratically) chose to go with –restoring– as we found it to be the better fitting term to describe the actual process. For this reason we would prefer to keep –restoring–.

â€¢ GlcNAc scan- â€œWhere Re-Glyco will try to fit a single GlcNAc monosaccharide into all the NXS/T sequons in the proteinâ€ â€œwill not the full N-linked structures also affect which sequon gets glycosylated? I accept that 93% of the structures you tried were correctly annotated butâ€!?

From the protein folding process, absolutely yes, but for the reconstruction of a glycoprotein from a folded template, not necessarily. The success rate of GlcNAc Scanning against experiment stems from the following. As the N-glycan is transferred co-translationally to the sequon as a -Man9-(Glc)3 in the unfolded protein, folding occurs around it, with the glycan left exposed on the surface and functionalized progressively depending on its degree of accessibility. When we start from an already folded protein with Re-Glyco, the minimum requirement for the N-glycan to be present is that the GlcNAc at its reducing-end spatially fits in the glycosylation site. That N-glycan is unlikely to be smaller than a paucimannose, and even if so, the level of degradation would indicate broad accessibility, so any N-glycan may fit.

As Rev.3 suggests, fitting of a whole N-glycan may not be possible for all degrees of functionalization at every occupied site, eg. a Man5 may fit well, but not a large, multiantennary complex N-glycan, and Re-Glyco is already extremely good at reporting such information. So, for sites whose occupancy is verified by GlcNAc Scanning, we are currently designing a tool that analyses on the fly the fit of different N-glycans types. The results of this analysis will lead us to generate potential glycosylation profiles for different cell lines that will inform the choice of glycosylation in the absence of an experimental profile. We are working on this in collaboration with experimentalists to build a ML algorithm to fulfil this task, while adding a more sophisticated potential energy function to Re-Glyco. While the development of this tool is beyond the scope of the present work and will entail a specifically aimed effort both computationally and experimentally, we believe that GlcNAc Scanning as it stands will be more broadly useful to the wider community and will remain the default functionality in Re-Glyco, with an Advanced Glycan-Profiling layer, for the expert users, when that will become available.

â€¢ A demonstration page would be useful on the webpage where the user is stepped through a real example â€œespecially if the demo shows how to add your own structures to different sites â€œUniProt does not have all these possibilities as you have said!

Absolutely and accordingly we have made and added short tutorial videos to the website (see tab â€œTutorialsâ€) to help users navigate different functionalities.

â€¢ Similarly in Sugar Drawer the â€œload Structureâ€ tool is cool â€œthough it wasnâ€™t clear to me how to add a single monosaccharide in a specific linkage â€œI eventually worked this out but again a demonstration example would be good?

11

Sugar Drawer is a tool developed by Prof Kinoshitaâ€™s group in Soka University. We added a link to the tutorial they produced.

â€¢ Need to define â€œclustersâ€ for the uninitiated as well as examples of what the plots on the structure pages are telling us?

We added to the FAQ more information to aid the users less familiar with statistical analysis. While a detailed description of the approach is described in Section 3 of the SI in the context of the Glycan Analysis Pipeline (GAP) for the more advanced/interested readers.

Overall the GlycoShape development is definitely needed for the field.

Thank you very much!

What I would say however is that I think the developers should work closely with an experimental group who trials the software and reports back on the pluses and minuses of the tool. This will ensure that it works in the hands of the less computationally informed and serves the experimental /biological community as well as it should.

Indeed, and this has been our goal throughout. Before the release its on Dec 11th, 2023, of GlycoShape has gone through extensive testing (2 months) by more than 25 colleagues, with expertise in structural biology and glycobiology, as well as glycoproteomics and

glycoanalytics, bioinformatics, biophysics, chemical biology and carbohydrate chemistry. As Rev.3, we do also believe that the development of a tool for the broad scientific community should be built on the feedback from the wider possible audience. We will continue to work this way for all future developments of GlycoShape.

12

Version 2:

Decision Letter:

Our ref: NMETH-A55080B

10th May 2024

Dear Elisa,

Thank you for submitting your revised manuscript "Restoring Protein Glycosylation with GlycoShape" (NMETH-A55080B). It has now been seen by the original referees and their comments are below. The reviewers find that the paper has improved in revision, and therefore we'll be happy in principle to publish it in Nature Methods, pending revisions to satisfy the referees' final requests (we would like Reviewer #2's requests to be addressed in the final version with appropriate wording changes, additional discussion where needed, and inclusion of relevant citations) and to comply with our editorial and formatting guidelines.

TRANSPARENT PEER REVIEW

Please note: we allow redactions to authors' rebuttal and reviewer comments in the interest of confidentiality. If you are concerned about the release of confidential data, please let us know specifically what information you would like to have removed. Please note that we cannot incorporate redactions for any other reasons. Reviewer names will be published in the peer review files if the reviewer signed the comments to authors, or if reviewers explicitly agree to release their name. For more information, please refer to our <https://www.nature.com/documents/nr-transparent-peer-review.pdf> target="new">FAQ page.

ORCID

Sincerely,
Arunima

Arunima Singh, Ph.D.
Senior Editor
Nature Methods

Reviewer #1 (Remarks to the Author):

The Authors have addressed all the points that I previously raised.

I commend the Authors for their excellent work, which is now ready for publication.

Reviewer #2 (Remarks to the Author):

The authors have responded extensively to the comments of all reviewers, and have made some minor modifications to the manuscript. In fact, in view of the very constructive comments of all three reviewers, I am somewhat surprised about the vehemence of the response by the authors and the very minor changes that were made to the manuscript to reflect these suggestions for improvements.

I was reviewer 2 of the previous submission and would like to go through my original comments:

1. Thank you for clarifying this.
2. I raised this point (and the next one) to encourage the authors to tone down statements that the aglycosylated protein structure is 'wrong'. They may be inappropriate representatives of the physiological relevant protein, but I maintain that the very local structure of the glycosylation site in an aglycosylated protein may very well be correctly solved experimentally and look different from the one in the glycosylated protein. These are two different proteins (glycosylated and non-glycosylated) and they can both be correct. I find statements that the structures are erroneous or wrong too harsh. I am surprised that the authors write in total a one page response to these points and did not consider to replace adjectives like 'wrong' with 'inappropriate for glycosylation'. It would not do any harm to their manuscript or their great work.
3. See the previous point.
4. The authors give an explanation why their approach worked well, but do not mention if they even considered an energy minimization. More importantly, they do not consider that other readers of the manuscript may have the same question and do not address this point in the manuscript.
5. The authors give a 1.5 page response highlighting why they consider their work radically different. I am surprised that they did not just acknowledge previous developments in their manuscript (CharmGUI, GlyTouCan, GlycoShield, the Turupcu reference). It doesn't diminish their work to acknowledge that others have done similar things before. With respect to the latter work, I disagree with the view of the authors that it is 'radically different': look at figure 9 of the indicated publication: pre-equilibrated ensembles of glycans are grafted onto the protein, and either the best fitting or all reasonable ones are selected. Whether these pre-equilibrated ensembles are generated from enhanced sampling or from vanilla MD is irrelevant.
6. OK, the authors explained their reasoning. Again, it would be thinkable that other readers have the same question, so their argumentation could be described more explicitly in the manuscript.

Reviewer #2 (Remarks on code availability):

For the revision, I did not re-evaluate the code or the website. I did not find a response from the authors to my previous evaluation.

Reviewer #3 (Remarks to the Author):

As recommended by all three reviewers in the first review - this paper should definitely be published.

Now that all comments have been answered and appropriate changes made to the manuscript along with promise of further future developments I have no hesitation in recommending publication now without further delay.

Reviewer #3 (Remarks on code availability):

I am not a bioinformatician so there is no point in my accessing the code. The other 2 reviewers would be able to comment on this aspect.

Author Rebuttal letter:

Please note that the reviewers's comments are reported verbatim in italics. Our answers are shown in normal text, highlighted in yellow.

Reviewer #2

Remarks to the Author:

The authors have responded extensively to the comments of all reviewers, and have made some minor modifications to the manuscript. In fact, in view of the very constructive comments of all three reviewers, I am somewhat surprised about the vehemence of the response by the authors and the very minor changes that were made to the manuscript to reflect these suggestions for improvements.

Apologies for the confusion on our behalf. We read most of the reviewers's comments as questions and curiosities that did not necessarily require integration into the manuscript. In fact, we found a good few queries quite interesting and enjoyed elaborating on the answer to possibly cover all possible angles the reviewer may have found interesting. Our answers

were indeed comprehensive and more suitable for a review, rather than a research article, so we chose not to incorporate them in the manuscript. Nevertheless, we will explicitly address all remaining queries from Rev.2 below in the new version of the manuscript.

I was reviewer 2 of the previous submission and would like to go through my original comments:

1. Thank you for clarifying this.

2. I raised this point (and the next one) to encourage the authors to tone down statements that the aglycolylated protein structure is 'wrong'. They may be inappropriate representatives of the physiological relevant protein, but I maintain that the very local structure of the glycosylation site in an aglycosylated protein may very well be correctly solved experimentally and look different from the one in the glycosylated protein. These are two different proteins (glycosylated and non-glycosylated) and they can both be correct. I find statements that the structures are erroneous or wrong too harsh. I am surprised that the authors write in total a one page response to these points and did not consider to replace adjectives like 'wrong' with 'inappropriate for glycosylation'. It would not do any harm to their manuscript or their great work.

Sincere apologies for this oversight. Rev. 2 is entirely correct here. We honestly missed a few of those definitions in our list of corrections, while meaning to replace the adjective 'wrong' everywhere. We have done that now. More specifically, on page 7 we replaced the definition 'few inaccuracies in the PDB 1E4J protein structure' with 'few features of the PDB 1E4J structure that make it incompatible with N-glycosylation'. On pages 7 and 8 we replaced the sentence 'the wrong orientation of the Asn sidechain, which in the specific case of N45 depends on the wrong assignment of the positions of ND2 vs. OD1, see Figure 4a.' with 'an orientation of the Asn sidechain that is unsuitable for N-glycosylation, which in the specific case of N45 depends on the assignment of the positions of ND2 vs. OD1, see Figure 4a'. On page 8 we replaced the sentence 'The correct orientation of the ND2 and OD1 at N45 (..)' with 'Indeed, in the presence of an N-glycan at N45 the orientation of the ND2 and OD1 is inverted (..)'. On page 8 we replaced the sentence 'As the correct assignment of coordinates to ND2 and OD1 is largely arbitrary (..)' with 'The assignment of coordinates to ND2 and OD1 is largely arbitrary in structures obtained from X-ray

1

crystallography (..)' Also on page 8 and throughout, we replaced the adjective 'wrong' with 'incompatible', which we agree with Rev.2 does not bear a negative connotation.

3. See the previous point.

See all corrections above.

4. The authors give an explanation why their approach worked well, but do not mention if they even considered an energy minimization. More importantly, they do not consider that other readers of the manuscript may have the same question and do not address this point in the manuscript.

In this case, we thought that the point above was a question and that the answer was not something that we needed to include in the manuscript. We have now addressed the matter by adding the following sentence on page 5 'The computational efficiency of this stochastic approach is vastly superior to any gradient-descent minimization algorithm, while providing comparable results'.

5. The authors give a 1.5 page response highlighting why they consider their work radically different. I am surprised that they did not just acknowledge previous developments in their manuscript (CharmGUI, GlyTouCan, GlycoShield, the Turupcu reference). It doesn't diminish their work to acknowledge that others have done similar things before.

The reason why we did not mention other computer-based methods to build complex carbohydrates, except for glycam.org that we use as a carbohydrate builder for the development of GlycoShape, is far from thinking it would detract from our work and we apologise if that was the impression this Reviewer got. We simply deemed it beyond the scope of a research article and more appropriate for a review article as those methods are quite different in terms of scope and applications. More specifically related to the methods this Reviewer mentioned above, CHARMM-GUI is a carbohydrate builder, very much like glycam.org, but developed (initially) for use in combination with the CHARMM software and all-atom additive force-field for MD simulations. Carbohydrate builders were not developed for structural glycobiology purposes on their own and thus are not related to the work we

presented. GlyTouCan is not a 3D structure database, but a 2D structure database, which is actually cited as Reference 28 as GlyTouCan IDs are assigned to all glycans in the GlycoShape GDB. Glycobiologists often refer to glycan sequences as "structures", so probably the confusion in terms of "structure repository" comes from there. Regarding GlycoShield and the method developed by Turupcu and Oostenbrink, we added the sentence on page 9 "(...) We believe that these features, in combination with its computational efficiency, make GlycoShape unique and distinct from computational tools such as GlycoShield69, designed to rapidly assess and quantify excluded volumes from glycan shielding, or from earlier work by Turupcu and Oostenbrink70 proposing the reconstruction of glycans through free-energy calculations."

With respect to the latter work, I disagree with the view of the authors that it is 'radically different': look at figure 9 of the indicated publication: pre-equilibrated ensembles of glycans are grafted onto the protein, and either the best fitting or all reasonable ones are selected.

2

Whether these pre-equilibrated ensembles are generated from enhanced sampling or from vanilla MD is irrelevant.

We respectfully disagree with the statement above. The ability of restoring accurate structural data depends on the quality of the MD sampling in combination with the ability to retain the structural information from experiment, especially where this information is very accurate (i.e. high resolution X-ray/cryo-EM), such as the rotameric conformation of the aglycone Asn or Ser/Thr. GlycoShield, at least at the moment, cannot provide either of those due to the limited sampling and because of how the glycans are linked to the protein, see below.

6. OK, the authors explained their reasoning. Again, it would be thinkable that other readers have the same question, so their argumentation could be described more explicitly in the manuscript.

We agree and to this end we added the following sentence on page 7 "Within this framework, the ability to retain structural information as default is crucial for the correct reconstruction of the glycosylation site. This is especially important for structural data obtained at high resolution, where the relative orientation of Asn sidechains, or of Ser/Thr sidechains for O-glycans, can be unequivocally assigned. Our approach is designed to preserve this information, which allows to restore the correct orientation of the glycan relative to the protein."

With this we would like to thank this Reviewer again for their careful reading of our work and for the useful feedback. We hope that the answers we provided to the queries and the modifications to the manuscript are satisfactory.

3
